# Interpreting pathways to discover cancer driver genes with Moonlight

Antonio Colaprico[1,2,3,19]*, Catharina Olsen[1,2,4,5,19], Matthew H. Bailey[6,7], Gabriel J. Odom [3,8], Thilde Terkelsen[9], Tiago C. Silva [3,10], André V. Olsen [9], Laura Cantini [11,12,13,14], Andrei Zinovyev [11,12,13], Emmanuel Barillot [11,12,13], Houtan Noushmehr[10,15], Gloria Bertoli [16], Isabella Castiglioni[16], Claudia Cava [16], Gianluca Bontempi[1,2,20], Xi Steven Chen[3,17,20]* & Elena Papaleo[9,18,20]*

Cancer driver gene alterations influence cancer development, occurring in oncogenes, tumor suppressors, and dual role genes. Discovering dual role cancer genes is difficult because of their elusive context-dependent behavior. We define oncogenic mediators as genes controlling biological processes. With them, we classify cancer driver genes, unveiling their roles in cancer mechanisms. To this end, we present Moonlight, a tool that incorporates multiple-omics data to identify critical cancer driver genes. With Moonlight, we analyze 8000+ tumor samples from 18 cancer types, discovering 3310 oncogenic mediators, 151 having dual roles. By incorporating additional data (amplification, mutation, DNA methylation, chromatin accessibility), we reveal 1000+ cancer driver genes, corroborating known molecular mechanisms. Additionally, we confirm critical cancer driver genes by analysing cell-line datasets. We discover inactivation of tumor suppressors in intron regions and that tissue type and subtype indicate dual role status. These findings help explain tumor heterogeneity and could guide therapeutic decisions.

[1] Interuniversity Institute of Bioinformatics in Brussels (IB)2, Brussels, Belgium. [2] Machine Learning Group, Université Libre de Bruxelles (ULB), Brussels, Belgium. [3] Department of Public Health Sciences, University of Miami, Miller School of Medicine, Miami, FL 33136, USA. [4] Center for Medical Genetics, Reproduction and Genetics, Reproduction Genetics and Regenerative Medicine, Vrije Universiteit Brussel, UZ Brussel, Laarbeeklaan 101, 1090 Brussels, Belgium. [5] Brussels Interuniversity Genomics High Throughput core (BRIGHTcore), VUB-ULB, Laarbeeklaan 101, 1090 Brussels, Belgium. [6] Division of Oncology, Department of Medicine, Washington University in St. Louis, St. Louis, MO 63110, USA. [7] McDonnell Genome Institute, Washington University, St. Louis, MO 63108, USA. [8] Department of Biostatistics, Stempel College of Public Health, Florida International University, Miami, FL 33199, USA. [9] Computational Biology Laboratory, and Center for Autophagy, Recycling and Disease, Danish Cancer Society Research Center, Strandboulevarden 49, 2100 Copenhagen, Denmark. [10] Department of Genetics, Ribeirão Preto Medical School, University of Sao Paulo, Ribeirão Preto, Brazil. [11] Institut Curie, 26 rue d'Ulm, F-75248 Paris, France. [12] INSERM, U900, Paris F-75248, France. [13] Mines ParisTech, Fontainebleau F-77300, France. [14] Computational Systems Biology Team, Institut de Biologie de l'Ecole Normale Supérieure, CNRS UMR8197, INSERM U1024, Ecole Normale Supérieure, Paris Sciences et Lettres Research University, 75005 Paris, France. [15] Department of Neurosurgery, Brain Tumor Center, Henry Ford Health System, Detroit, MI, USA. [16] Institute of Molecular Bioimaging and Physiology of the National Research Council (IBFM-CNR), Milan, Italy. [17] Sylvester Comprehensive Cancer Center, University of Miami Miller School of Medicine, Miami, FL 33136, USA. [18] Translational Disease System Biology, Faculty of Health and Medical Science, Novo Nordisk Foundation Center for Protein Research, University of Copenhagen, Copenhagen, Denmark. [19] These authors contributed equally: Antonio Colaprico, Catharina Olsen. [20] These authors jointly supervised this work: Gianluca Bontempi, Xi Steven Chen, Elena Papaleo. *email: axc1833@med.miami.edu; steven.chen@med.miami.edu; elenap@cancer.dk

Cancer is a complex and heterogeneous disease, hallmarked by the poor regulation of critical functions, such as growth, proliferation, and cell-death pathways. To better understand the hallmarks of cancer, such as proliferation and apoptosis, it is critical to accurately identify cancer driver genes. Due to a strong dependency on the biological context, cancer driver genes and their roles in specific tissues are elusive to annotate, and their discovery is often complicated. In a recent review, cancer progression was summarized across four different steps: cancer initiation, tumor propagation, metastasis to distant organs, and drug resistance to chemotherapy[1]. Cancer progression is accelerated by the accumulation of genomic abnormalities in two different categories of cancer driver genes: oncogenes or tumor suppressors[2]. The gain-of-function of oncogenes together with the loss-of-function of tumor suppressors determine the processes that control tumor formation and development[3].

Certain cancer driver genes can exhibit oncogene or tumor-suppressor behavior depending on the biological context, which makes them difficult to identify. We will call such genes dual-role cancer driver genes[4,5]. A motivating example for our study is the dual-role gene NOTCH. This gene is considered a hematopoietic proto-oncogene in T-cell acute lymphoblastic leukemia, while it has a tumor-suppressor role in solid tumors—such as basal cell carcinoma of the skin, hepatocellular carcinoma, and in some forms of leukemia[6]. In addition, it has been shown that concomitant Notch activation and p53 deletion trigger epithelial-to-mesenchymal transition and metastasis[7].

Recently, TCGA Pan-Cancer Atlas Initiative[8] amassed findings into a suite of 27 studies covering 11,000 tumors from 33 of the most frequent types of cancers[9–11]. These studies investigated cancer complexity from different angles and integrated different sources of -omics data (i.e., gene, protein, and microRNA expression, somatic mutations, DNA methylation, copy-number alterations, and clinical data). In particular, this initiative employed many computational tools to identify 299 cancer driver genes and >3400 driver mutations[12]. Although these methods were demonstrated to be effective, it remains fundamental to clarify the role of cancer driver genes, inspect the consequences of cancer alterations, and link the identified patterns with the underlying biological effects.

Several approaches have been developed to discover cancer driver genes and pathways, but these methods did not harness the power of integrating biological processes and their connection with gene deregulation to predict cancer driver genes[12]. Our approach allows the interpretation of cancer-related pathways to identify essential cancer driver genes by integrating information on biological processes from literature with gene–gene interactions in transcriptomic data. This approach unlocks the possibility of identifying context-dependent cancer genes. We then prioritize genes discovered by Moonlight according to the analysis of additional multi-omics data. If the gene exhibits significant evidence after additional data integration, we define the genes that Moonlight discovered as cancer driver genes. Moreover, investigating the intra- and inter-tumor heterogeneity, we identified dual-role genes within cancer types or subtypes.

## Results

**Overview of Moonlight**. We here present Moonlight: a tool designed to identify cancer driver genes that moonlight as opposite roles when observed in the context of transcriptomic networks. The name refers to (i) the concept of protein moonlighting (or gene sharing) is a phenomenon by which a protein can perform more than one function[13], and (ii) casting genes in a new light can lead to improved treatment regimens and prognostic indicators.

Moonlight can detect cancer driver-gene events specific to the tumor and tissue of origin, including potential dual-role genes, as well as elucidate their downstream impact. To accomplish this, Moonlight integrates information from literature, pathways databases, and multiple -omics data into a comprehensive assessment of a gene's role and function (Fig. 1a). Moonlight is freely available as an open-source R package within the

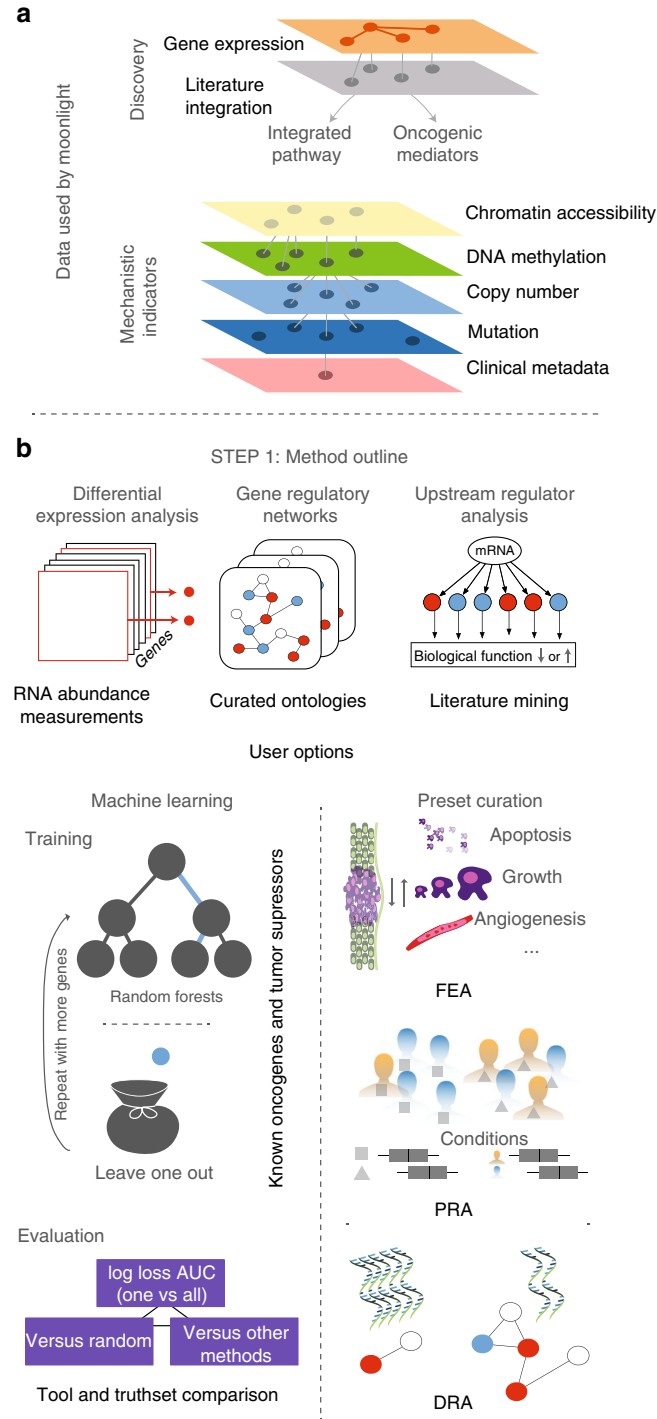

**Fig. 1 Moonlight data integration and functionalities. a** Data used for discovery of oncogenic mediators and controlling mechanisms of cancer driver genes. **b** Moonlight pipeline for discovery of tumor suppressors, oncogenes, and dual-role genes.

Bioconductor project at http://bioconductor.org/packages/MoonlightR/.

The main concept behind Moonlight relies on the observation that the classical approach to experimentally validated cancer driver genes consists in the modulation of their expression in cellular assays, together with the quantification of process markers, such as cellular proliferation, apoptosis, and invasion. We thus selected apoptosis and cell proliferation as main gene programs to detect cancer driver genes. To accomplish this task, we manually curated over 100 biological processes linked to cancer, including proliferation and apoptosis. During this manual curation, we gave Moonlight information on whether the activation of each process leads to promotion or reduction of cancer (Methods; Supplementary Data 1 and 2). Once Moonlight identifies an oncogenic process altered in tumors using gene expression data, it detects genes that activate or inhibit this process. We define such genes as oncogenic mediators. Oncogenic mediators in bulk-tumor samples and cell-line experiments that are also co-explained by other factors, such as DNA methylation, copy number, clinical data, drug–target, or chromatin accessibility, are retained in the analyses.

The rationale behind this two-step process is that gene expression alone may lead to a large number of candidate genes that are not necessary driving the cancer phenotype. A second layer of evidence is necessary for a cancer driver gene to be activated and promote a cancer phenotype. Therefore, Moonlight explores the oncogenic mediators detected by gene expression, and when Moonlight identifies a second evidence (such as hyper- or hypomethylation), we predict that the oncogenic mediators can be defined as critical cancer driver genes. Therefore, the prediction of cancer driver genes can be achieved using the integration of gene expression and prioritization of biological process mediators using multiple data types.

Moonlight offers two approaches: expert- and machine learning. While both of these approaches identify cancer driver genes using gene expression data as a major source of information (Fig. 1b; Methods), the expert-based approach offers the potential to incorporate user expertise to reveal otherwise hidden molecular mechanisms used by cancer driver genes.

**Moonlight identifies oncogenic mediators in breast cancer**. In the first application of Moonlight, we employed the expert-based approach and selected apoptosis and cell proliferation as the representative biological processes, studying 18 cancer types from TCGA (Methods). We compared tumor and normal samples using sample profiles from multiple -omics data retrieved from the Genomic Data Commons using the TCGAbiolinks[14] package and a workflow that we developed to process cancer data[15,16] (Methods; Supplementary Data 3). Specifically, we selected breast-invasive carcinoma from TCGA for illustrative purposes. In this step of the analysis, we found 3390 genes that were differentially expressed (Methods, Supplementary Data 3) when comparing normal and tumor breast-cancer tissue samples. Functional Enrichment Analysis (Methods) revealed that these genes were significantly enriched in 32 biological processes (Fig. 2a; Supplementary Data 4). Several biological processes promoting cancer progression (cell proliferation, invasion of cells, inflammatory response) were significantly increased. Concurrently, processes counteracting cancer progression (branching of cells, apoptosis of tumor cell lines) were significantly decreased.

One example of a biological process associated with cancer progression is increased cell proliferation. The cell proliferation biological process, as defined by Gene Ontology and KEGG database, has 3938 annotated genes, of which 1172 were identified by Moonlight to be differentially expressed genes (Student's $t$ test FDR-adjusted $p = 4.38E-113$) (Fig. 2a; Supplementary Data 4, Methods). Another example is apoptosis, which is generally downregulated in association with cancer progression. This process had 1284 annotated genes, of which 390 were found to be differentially expressed (Student's $t$ test FDR-adjusted $p = 3.15E-34$) (Fig. 2a; Supplementary Data 4). Moonlight identified a significant decrease of apoptosis in the comparison of tumor versus normal samples. Overall, Moonlight predicted 776 cancer driver genes (626 oncogenes and 150 tumor suppressors) in the analyses of breast-invasive carcinoma (Supplementary Data 5).

We also showed the ability of Moonlight to identify associations between the aforementioned biological processes and the specific genes that regulate these processes. To accomplish this, we performed Pattern Regulation Analysis (Methods), enabling the identification of genes with two distinct patterns. These patterns (Fig. 2b) were (i) increased proliferation and decreased apoptosis (e.g., CDC20[17], TIMELESS[18], and CDC6[19]), and (ii) decreased proliferation and increased apoptosis (e.g., ADAMTS9[20], DLL4[21], and SOX7[22]). We supported our findings by literature searches (Fig. 2b) and hypothesize that genes with pattern (i) can act as oncogenes while genes with pattern (ii) can act as tumor suppressors.

**Moonlight applied to pan-cancer data**. To illustrate its potential, we applied the Moonlight pipeline to contrast normal and tumor samples for 18 cancer types (Methods). Moonlight used apoptosis and cell proliferation as key markers to identify 3123 unique oncogenic mediators (Supplementary Data 6, Methods). We classified the genes that concurrently increased apoptosis and decreased proliferation as tumor-suppressor genes, and vice versa for oncogenes.

Of the 3123 oncogenic mediators within the comprehensive set of 18 cancer types, the Moonlight pipeline identified 1076 tumor-suppressor-like and 1896 oncogene-like mediators (Fig. 2c; Supplementary Data 6). In addition, 151 driver genes showed a dual-role effect (Fig. 2d; Supplementary Fig. 1, Supplementary Data 6). We have characterized the specific molecular changes associated with all the 3123 oncogenic mediators and cancer driver genes in the following sections.

**Cancer driver genes are associated with cancer heterogeneity**. Moonlight can be used to investigate cancer molecular subtypes, here illustrated using breast-cancer data. We compared normal breast tissue samples with samples from different molecular subtypes of breast cancer, according to the PAM50 classification[23]. This analysis revealed a total of 638 cancer driver genes specific to individual subtypes: luminal A (221 oncogenes and 180 tumor suppressors); luminal B (51 oncogenes and 73 tumor suppressors); basal-like (14 oncogenes and 76 tumor suppressors); HER2-enriched (8 oncogenes and 15 tumor suppressors) (Fig. 2e; Supplementary Data 5). In addition, Pattern Recognition Analysis combined with Dynamic Recognition Analysis (Supplementary Software 1) revealed several specific gene programs increased or decreased according to the specific molecular subtype of the cancer of study (Supplementary Fig. 2; Methods).

We identified FOXM1 as an oncogene in the luminal A subtype, a gene known to be a lineage-specific oncogene in this subtype[24]. The forkhead box (Fox) A1 and M1 genes belong to a superfamily of evolutionarily conserved transcriptional factors, and FOXM1 has been shown to be a promising candidate target in the treatment of breast cancer[25]. It is known that the binding of a transcription factor to the promoter region of a target gene is

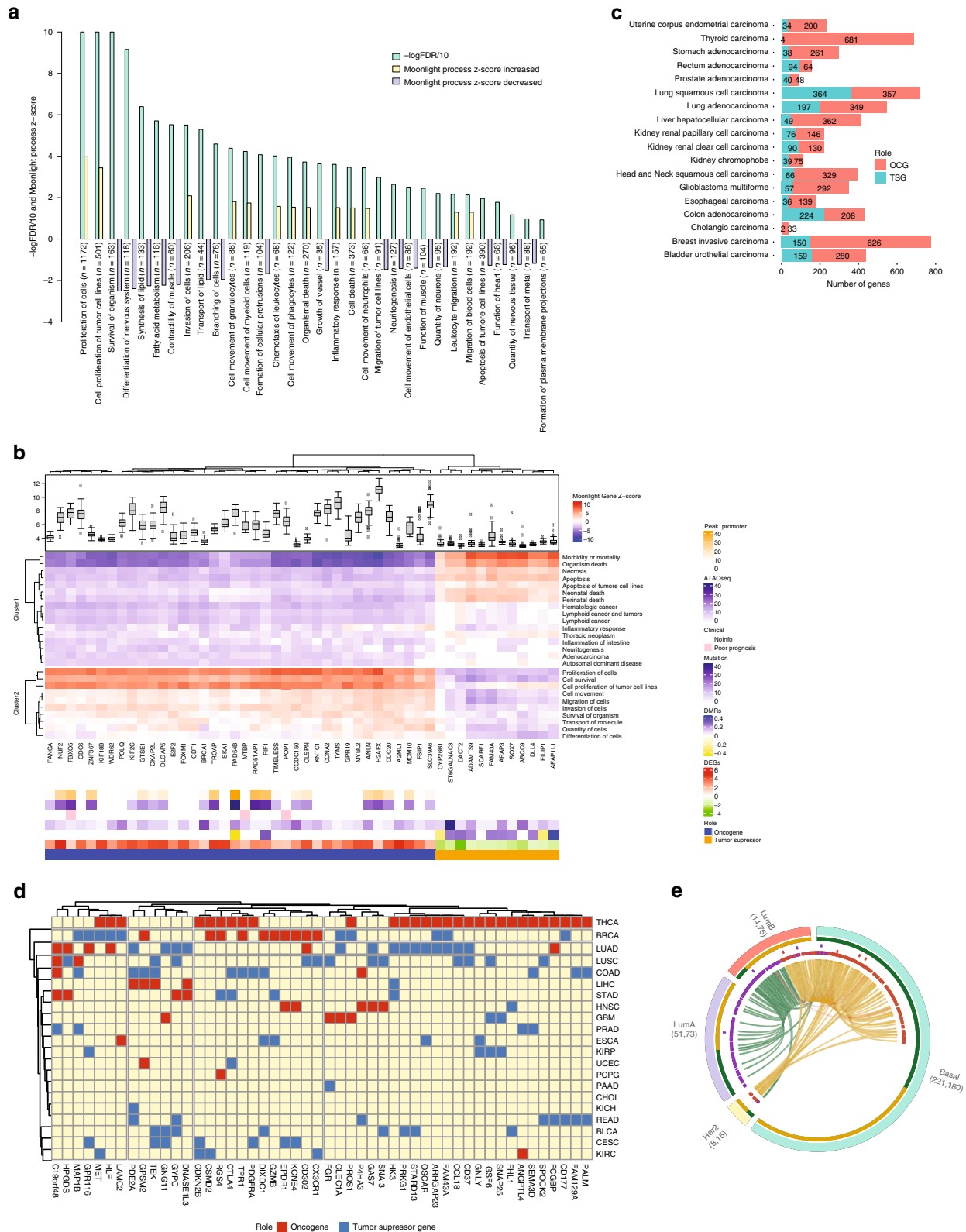

restricted by complex chromatin accessibility[26]. We looked at FOXA1 chromatin signal and we observed an association with open states of chromatin.

**DNA methylation controls activity in cancer driver genes.** To further investigate Moonlight findings, we explored additional

patterns using DNA methylation. In the literature, we observe the existence of two broad classes of CpG-methylated sites: (i) those with a strong inverse correlation between DNA methylation and chromatin accessibility across cell types and (ii) those with variable chromatin accessibility but constitutive hypomethylation[27]. Therefore, we identified differentially methylated regions between normal and tumor samples for 18 TCGA cancer types (Methods).

**Fig. 2 Moonlight application within breast-cancer case study. a** Barplot from Functional Enrichment Analysis showing the BPs enriched significantly with | Moonlight Process Z-score| > = 1 and FDR < = 0.01; increased levels are reported in yellow, decreased in purple, and green shows the -logFDR/10. A negative Moonlight Process Z-score indicates that the process' activity is decreased, while a positive Moonlight Process Z-score indicates that the process' activity is increased. Values in parentheses indicate the number of genes in common between the genes annotated in the biological process and the genes used as input for the functional enrichment. **b** Heatmap showing the top 50 predicted tumor suppressors and oncogenes in breast cancer and their associated biological processes. Hierarchical clustering was performed on the Euclidean distance matrix. Biological Processes with increased (decreased) Moonlight Gene Z-score are marked in red (blue). The number of samples reporting the mutation of specific genes ranges from white to dark purple. Hypermethylated (hypomethylated) DMR are shown in blue (yellow). Genes with poor Kaplan–Meier survival prognosis are marked in pink. Chromatin accessibility in the promoter region ranges from white (closed) to orange (open). The upper panel shows boxplots of cell-line expression levels. **c** Barplot reporting the number of tumor-suppressor genes (blue) or oncogene (red) predicted in pan-cancer analysis using expert knowledge paired with PRA using two selected biological processes, such as apoptosis and cell proliferation. **d** Heatmap showing the top 50 dual-role genes (by Moonlight Gene Z-score) within cancer types, oncogenes (OCGs) are shown in red and Tumor-Suppressor Genes (TSGs) in blue. TCGA study abbreviations available at https://gdc. cancer.gov/resources-tcga-users/tcga-code-tables/tcga-study-abbreviations. **e** Circos plots for molecular subtypes of Moonlight genes predicted using expert knowledge paired with PRA using two selected BPs, such as apoptosis and cell proliferation. From outer to the inner layer, the color labels are breast-cancer subtype. In the parentheses, the number of OCGs and TSG for a specific molecular subtype; OCGs (green) and TSGs (yellow); purple and orange for mutations: inframe deletion, inframe insertion, missense; gene–gene edges between two cancer molecular subtypes are OCG in both (green), TSG in both (yellow), dual-role genes (red).

Using Moonlight's expert-based approach, we integrated RNA and epigenetic data to identify critical genes.

Among 3310 oncogenic mediators in 18 cancer types, we saw that 1176 depicted epigenetic changes (509 oncogene-like, 586 tumor-suppressor like). Moonlight detected 233 genes associated with hypermethylation (tumor-suppressor critical) and 404 with hypomethylation (oncogene critical). We considered these genes to be critical epigenetic cancer driver genes. Among these genes, 18 cancer driver genes showed a dual role associated with epigenetic changes (Supplementary Data 7), five of which were considered to be critical: SLC27A6, PDGFRA, GAS7, PLXNC1, and NRP2. For example, Moonlight identified GAS7 as a hypermethylated tumor suppressor in lung cancer and as an hypomethylated oncogene in head-and-neck squamous cell tumors. These findings were confirmed by data on lung cancer[28], and associated with copy-number changes in head-and-neck cancer cell lines[29], but it has not been validated yet as oncogene for head-and-neck tumors, suggesting an interesting target for future studies.

For breast cancer, we found that 231 (30%) of the predicted oncogenic mediators experienced epigenetic changes. Of these genes, 54 tumor suppressors showed hypermethylation while 80 oncogenes showed hypomethylation. We considered these 134 genes to be critical epigenetic cancer driver genes for breast cancer. We inspected the 50 cancer driver genes for breast cancer with the highest Moonlight Gene Z-scores (Methods), of which Moonlight identified 14 tumor suppressors (Fig. 2b). Of these, eight reported hypermethylation in tumor samples (including ADAMTS9, DLL4, and SOX7, described above), while CYP26B1 and FILIP1 reported hypomethylation (Supplementary Data 7). ADAMTS9 exhibited promoter hypermethylation and its down-regulation is associated with decreased cell proliferation and increased apoptosis. Interestingly, these findings were confirmed by a recent study[20].

Among the cancer driver genes that experienced epigenetic changes in at least five cancer types, we identified eight genes: CEP55, PIF1, RRM2, NCAPH, ZEB2, CIT, FLI1, and PCDH17. Moonlight detected RRM2 as an oncogene. This gene is a critical epigenetic cancer driver gene (hypomethylated) in six cancer types, including head-and-neck and lung cancer, and is associated with multiple other cancers. Recently, it was shown that knockdown of RRM2 led to intrinsic apoptosis in head-and-neck squamous cell carcinoma and non-small cell lung cancer cell lines, confirming our findings[30].

In addition, Moonlight identified FLI1 as a tumor suppressor in multiple cancer types, including lung, breast, uterine, and colon (Supplementary Data 7). We also found hypermethylation of colon adenocarcinoma and lung adenocarcinoma, specifically in two CpG loci associated with FLI1: cg11017065 (colon cancer) and cg04691908 (lung adenocarcinoma). We hypothesize that differentially methylated CpG islands, or hypermethylation of the FLI1 promoter, may also lead to inactivation of FLI1's tumor-suppressor ability. FLI1 is known to be downregulated in colon adenocarcinomas and is associated with colon cancer progression[31]. Hypermethylation, especially in tumor suppressors, is a well-known epigenetic control mechanism that is important for gene inactivation in cancer cells[32]. Furthermore, DNA hypomethylation can be found early in carcinogenesis, and is often associated with tumor progression and oncogenes[33].

Therefore, Moonlight's highlighted mechanisms on CpG-island promoter regions can be summarized as follows: (i) oncogene activation is associated with DNA hypomethylation at the promoter sites, and (ii) tumor-suppressor inactivation is associated with DNA hypermethylation at the promoter sites. In general, epigenetic changes in promoter regions influence the activation of oncogenes and inactivation of tumor suppressors, but genes that have pre-existing sites for initiation of transcription with open chromatin are more likely to be activated after nuclear transfer[34]. This suggests that the chromatin signature influences transcriptional reprogramming, in which activated genes associated with new open chromatin sites—especially in transcription factors—play an important role.

**Cancer driver genes are prioritized at accessible regions.** Because epigenetic changes cooperate with chromatin accessibility to influence transcriptional activities, we also investigated if cancer driver genes predicted by Moonlight showed molecular changes at the level of chromatin accessibility. We performed integrative analysis of gene expression and ATAC-seq data on the 18 TCGA cancer types selected for our study. We detected five cancer types (breast-invasive carcinoma, glioblastoma multiforme, liver hepatocellular carcinoma, lung adenocarcinoma, lung squamous cell carcinoma) that showed higher chromatin accessibility peak signals in promoter regions for oncogenes than tumor suppressors, as predicted by Moonlight (Student's $t$ test $p < 0.05$, Fig. 3a). In contrast, the tumor suppressors showed higher peaks in intron regions compared with the oncogenes in six cancer types (Student's $t$ test $p < 0.05$, Fig. 3b). Interestingly, these results were mutually exclusive: the six cancer types with higher peaks at the intron regions for tumor suppressors did not show significant peaks in the promoter regions for oncogenes (Supplementary Data 8, Methods).

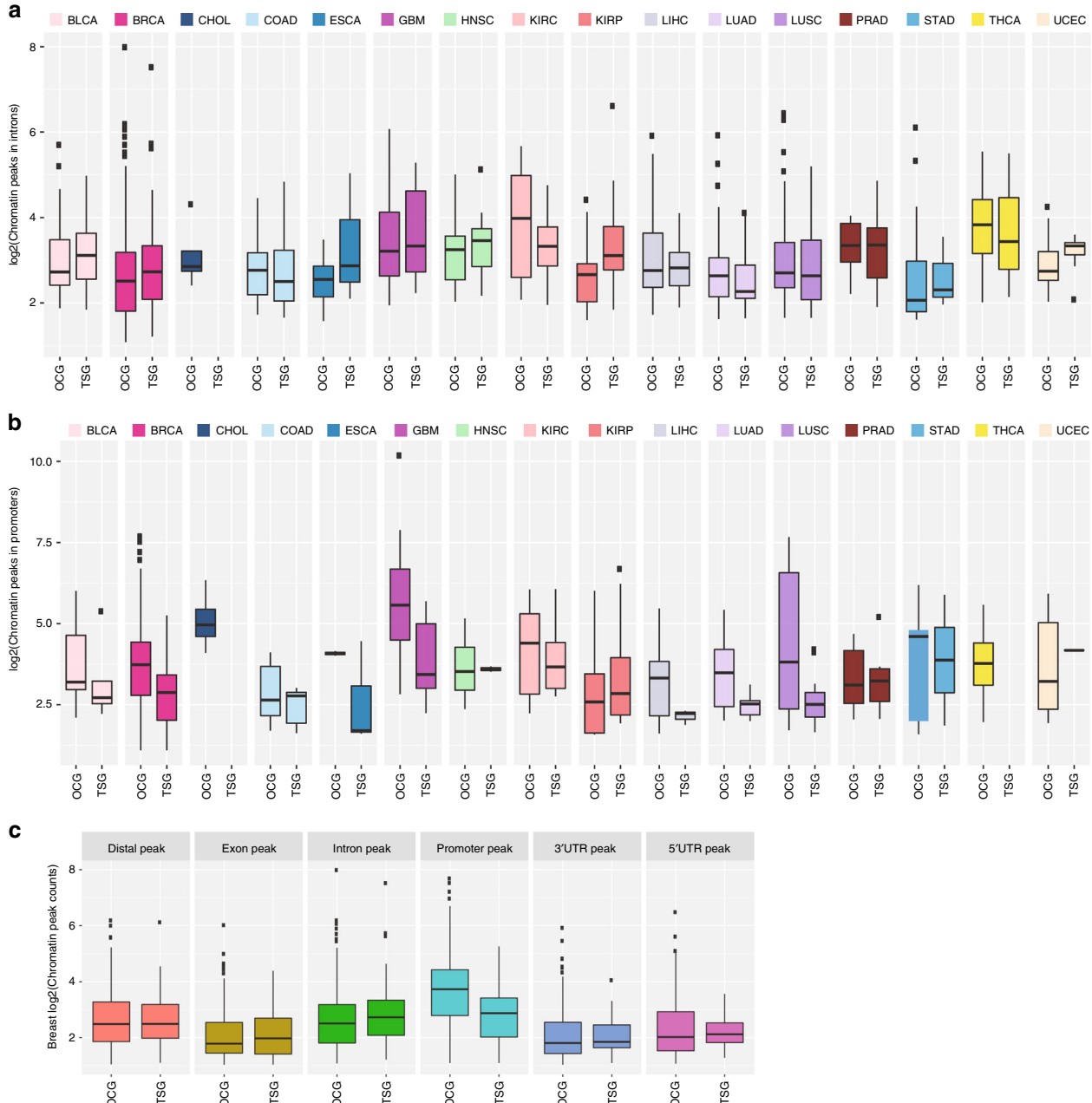

**Fig. 3 Chromatin accessibility landscape of oncogenic mediators. a** log2 (chromatin peaks in promoters) for tumor suppressor and oncogenes detected in Pan-Cancer study, **b** boxplot showing log2 (chromatin peaks in introns), and **c** breast cancer log2 (chromatin peaks count).

Moonlight identified mutually exclusive peaks in different regions: open chromatin in the intron region for tumor suppressors (Fig. 3b) and open chromatin in promoter regions for oncogenes (Fig. 3a). We also reported overall higher chromatin peaks signal for oncogenes when compared with tumor suppressors (Fig. 3c). Notably, LSM1, predicted by Moonlight as an oncogene and reported as an oncogene in breast cancer[35], showed the highest peak in the promoter region (followed by ERBB2, PSMD3, and PRR15). Supplementary Fig. 3a shows the PSMD3 peak signal for a selection of TCGA breast-invasive carcinoma ATAC-seq samples, while Supplementary Fig. 3b, c show the peak signals of ERBB2, PRR15, and GATA3. Moonlight identified the cell cycle kinase CDK4 as an oncogene in glioblastoma multiforme, with the highest normalized peak score (1164). Li et al. and Lubanska et al. reported that CDK4

inhibitor therapy was more effective in the glioblastoma proneural subtype[36,37].

In particular, among 151 dual-role genes detected by Moonlight one interesting gene, ANGPTL4, was predicted to be an oncogene in kidney cancers with associated promoter peaks as well as a tumor suppressor in prostate adenocarcinoma with hypermethylation in the promoter region (Supplementary Data 7, 8; Methods). Thus, Moonlight detected ANGPTL4 as a dual-role gene, a finding which was confirmed by a recent study[38].

A similar behavior was observed for SOX17, which was predicted as an oncogene in uterine corpus endometrial carcinoma associated with promoter peaks and as a tumor suppressor associated with hypermethylation in lung squamous cell carcinoma (Supplementary Data 7, 8; Methods) These findings were confirmed by ChipSeq of SOX17 in endometrial

cancer[39], while SOX17 suppressed cell proliferation and promoter hypermethylation has been shown in lung cancer[40].

**Critical cancer driver genes reshapes copy-number landscape.** The relationship between DNA hypomethylation of oncogenes, hypermethylation of tumor suppressors, and copy-number amplification or deletion is another well-known mechanism to modulate cancer driver genes[41]. We investigated if cancer driver genes predicted by Moonlight showed molecular changes at the copy-number level. For the 3123 mediators predicted by Moonlight within 18 cancer types, 848 showed copy-number changes and 358 showed critical copy-number cancer driver genes (eg. observed amplification of oncogenes and deletion of tumor suppressors) (Supplementary Data 9). For example, we observed amplification of the oncogenes CCND1 (supported by study[42]) and CCNE1 in breast cancer. Moreover, we identified deletions in tumor suppressors, such as DACT2 and TGFBR3 (Fig. 4a). In addition, Moonlight predicted FOXM1 as an oncogene with associated amplification in colon adenocarcinoma and lung squamous cell carcinoma[43,44]. Among the 151 predicted dual-role genes, 19 were identified with associated copy-number changes, while 12 genes were critical copy-number cancer driver genes, including ADAM6, BCL2, CACNA2D2, CDKN2B, CLEC1A, DIXDC1, FAM129A, GPSM2, IQGAP2, MAP1B, PALM, and TSPAN4. Moonlight predicted ADAM6, a dual-role lncRNA, as a novel tumor suppressor in colon cancer and oncogene in head-and-neck cancer.

Moonlight also showed that the anti-apoptotic BCL2 is a dual-role gene. Specifically, Moonlight identified BLC2 as an oncogene in thyroid carcinoma, through decreasing apoptosis and showing a peak in the exon region concurrently, confirmed by published data[45]. Moonlight also identified BCL2 as a tumor suppressor in prostate adenocarcinoma with promoter hypermethylation, deletion, and associated with increased apoptosis (Supplementary Data 7, 9). The BLC2 anti-apoptotic effect is a well-known mechanism in pancreatic cancer, especially because upregulation is required for pancreas progression, which implies that down-regulation can inhibit cancer progression.

**Oncogenic mediators exhibit differences in mutations.** Furthermore, we extended our study to mutation data. While it has been shown that highly mutated genes promote cancer progression[12], it is yet unknown if methylation and copy-number changes to cancer driver genes directly imply that these genes have been mutated. Therefore, we also investigated which cancer driver genes exhibited alterations at the mutational level. Moonlight applied to pan-cancer data revealed mutations in intron region (Fig. 4b) for tumor suppressors and mutations in promoter regions for oncogenes. (Fig. 4c). In Fig. 4d, we report the results of the analysis from different mutation types for the cancer driver genes predicted by Moonlight in breast cancer. Moonlight identified three oncogenes, CMYA5, ASPM, and ERBB2, showing 34, 30, and 29 samples with missense mutations, respectively (Methods; Supplementary Data 10). ASPM and CMYA5 are predicted as novel oncogenes in breast cancer, while ERBB2 is an already well-known oncogene in breast cancer[46]. Furthermore, ST6GALNAC3 was predicted by Moonlight to be a tumor suppressor in breast cancer with 33 samples with intron mutations. Therefore, we show the mutation site for the ST6GALNAC3 gene (Supplementary Fig. 4b).

Interestingly, Moonlight detected GATA3 as an oncogene in breast cancer with several mutated samples: frameshift insertion, deletion, and splice site. In particular, we observed that GATA3 showed the highest mutation rate in breast-cancer samples in splice-site and frameshift insertions. Therefore, we

show the mutation site (x308, D335, p408) for the GATA3 gene (Supplementary Fig. 4a). GATA3 is known to be an oncogene in breast cancer[47]. However, GATA3 has also been recently reported as a tumor suppressor for breast cancer in certain contexts[47], which intrigued us. In a recent study, we applied Moonlight to discover several pathways that are differentially expressed between wild-type GATA3 and GATA3 with frameshift/nonsense or missense mutations in breast-cancer samples[10]. GATA3-mutant cells are known to become more aggressive and exhibited faster tumor growth in vivo[48]. In this light, we believe that Moonlight was not only able to detect the oncogene behavior of GATA3 in breast cancer with precision but was also able to elucidate the underlying mechanism and mutation sites (Methods, Supplementary Fig. 4a).

**Oncogenic mediators impair survival outcomes.** It is well known that highly expressed oncogenes in cancer patients are associated with a worse prognosis[49], negatively impacting survival outcomes, whereas tumor suppressors present better outcomes[50,51]. With this in mind, we examined which oncogenic mediators could be associated with prognosis. Notably, an overall survival analysis identified 1051 prognostic cancer driver genes (Methods; Supplementary Data 11). Of these, 521 oncogenes were associated with poor prognosis, whereas 50 tumor suppressors with good prognosis. Interestingly, among these cancer driver genes, ADHFE1[52], TRPM8[53], and PGBD5[54] were not present in the gold-standard gene set from COSMIC and Vogelstein (Methods), but were recently validated as oncogenes for breast cancer[52–54]. Similarly, genes such as MTHFD2[55], CHAC1[56], and SDC1[57] were associated with poor prognosis in TCGA breast-cancer samples by Moonlight, and they were shown in literature to influence cell migration and proliferation in breast-cancer cell lines[56,58,59] (Supplementary Fig. 4c).

Subsequently, we explored the possibility that dual-role genes could differentially influence prognosis by cancer type or subtype. We examined the behavior of ANKRD23 (Ankyrin Repeat Domain 23). Moonlight predicted this gene to be an oncogene in renal clear-cell carcinoma associated with poor survival (log-rank test $p = 0.001$, Fig. 5a). Interestingly, Moonlight also predicted this gene to be a tumor suppressor in bladder urothelial carcinoma with good survival prognosis (log-rank test $p = 0.022$, Fig. 5b). Moonlight, applied in conjunction with clinical data, can highlight dual-role genes with variable impact on cancer survival across cancer types and subtypes.

**Moonlight machine-learning approach and tool comparison.** To show the second option of Moonlight, we applied the machine-learning approach to TCGA Pan-Cancer RNA-seq samples. We trained a random forest model on a gold-standard gene set of known cancer driver genes (Methods; Fig. 6a). We supplied the output of the Moonlight Upstream Regulatory Analysis (Methods) to this model to score the biological processes.

The machine-learning approach predicted four genes as candidate dual-role genes: BCL2, CDKN2A, KIT, and SOCS1 (Methods; Fig. 6b). Recent findings support the dual-role behavior of these four genes. BCL2's dual role is related not only to its expression but also to the localization of its protein products[60]. Also, for CDKN2A, the up- or downregulation of this gene has been described in several types of cancer, suggesting a dual role of the encoded protein[61]. Moreover, the cellular localization of the gene products (p15, p16, and p14ARF) appears to have different functions in different cancer types[62]. Furthermore, c-Kit's dual-role behavior in different contexts has been already proposed[63]. Finally, SOCS1 is known to act as a tumor

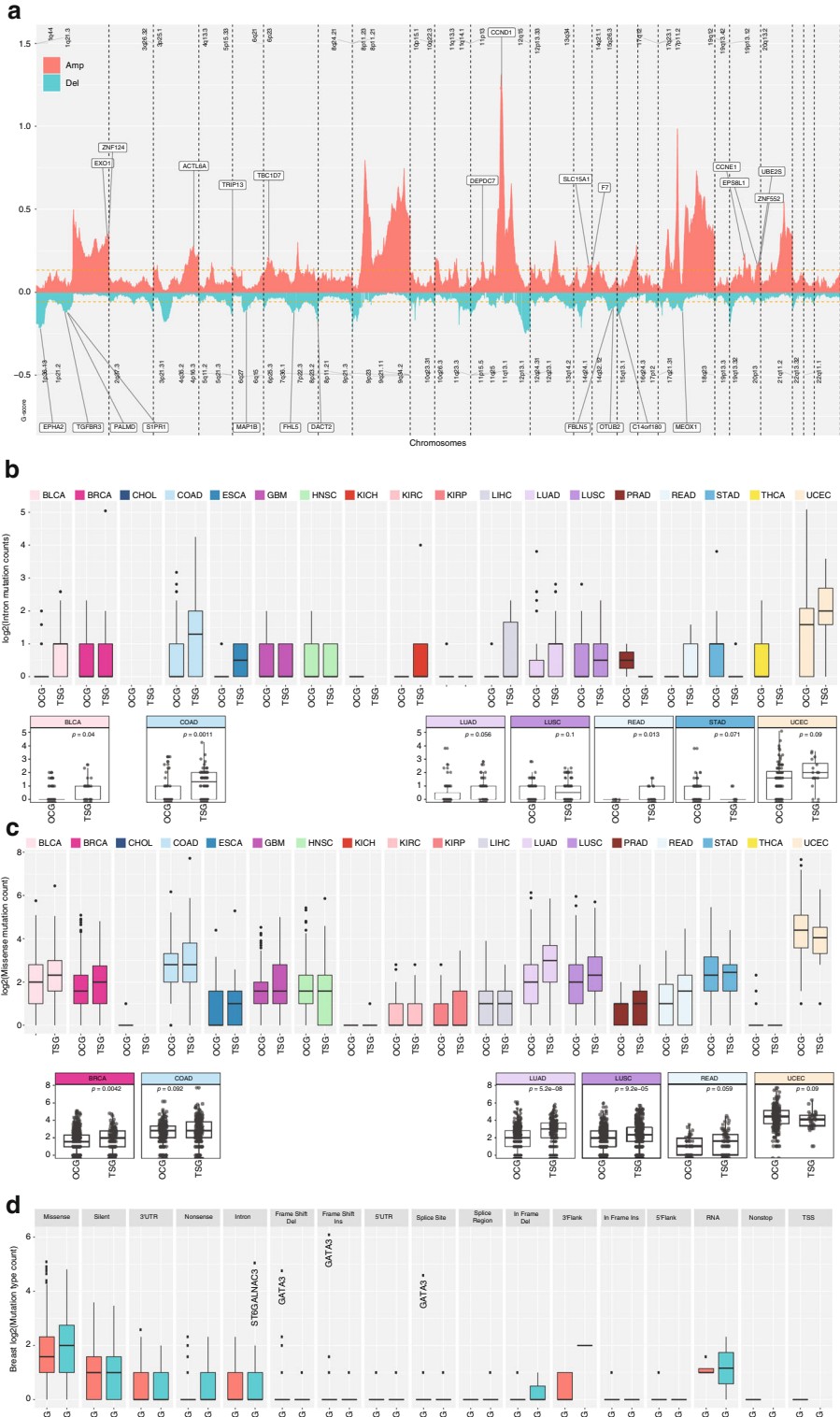

**Fig. 4 Copy number and mutational landscape of oncogenic mediators. a** Copy-number changes in breast cancer (amplification of oncogenes in red and deletion of tumor suppressors in blue) identified according to criteria described in the Methods section. The orange line represents the significance threshold (FDR = 0.25). The complete list of chromosome location peaks associated to cancer driver genes in Pan-Cancer study is included in Supplementary Data 8. **b** Boxplot showing log2 (intron mutation counts), **c** log2 (missense mutation counts) for tumor suppressor and oncogenes detected in Pan-Cancer study, and **d** breast cancer log2 (mutation type count).

suppressor in some cancer types[64] and as an oncogene in others[65].

To evaluate the performance of Moonlight, we compared its machine-learning approach to two state-of-the-art methods for

the detection of cancer driver genes: 20/20+[66] and OncodriveR-ole[67]. We chose these methods for their popularity, ease of implementation, and similarity to Moonlight's machine-learning approach. We conducted leave-one-out cross-validation for one

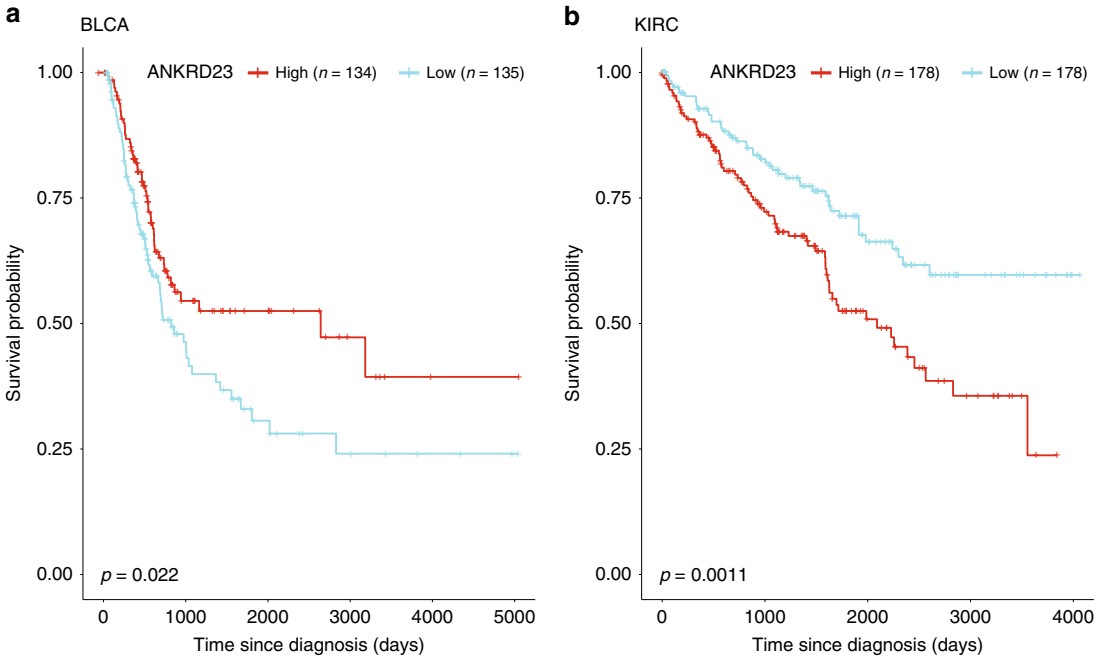

**Fig. 5 Moonlight dual-role genes that could differentially influence prognosis by cancer type or subtype.** Clinical implication (**a**, **b**) Kaplan–Meier survival curves show that ANKRD23 is a tumor suppressor in BLCA (**a**) and an oncogene in KIRC (**b**).

class versus all, and we found comparable results to Moonlight (Methods; Fig. 6c).

We observed that three cancer types obtained better performance (lowest log-loss values), namely esophageal carcinoma, kidney renal papillary cell carcinoma, and rectum adenocarcinoma (Fig. 6c), while liver hepatocellular carcinoma and head–neck squamous cell carcinoma had poorer performance. The discrepancies could be related to the source of oncogenes and tumor suppressors that we used to train and validate our model. The COSMIC and Vogelstein oncogene/tumor-suppressor lists (Methods) are not designed to be cancer specific. Therefore, it is likely that some of the oncogenes/tumor suppressors are not playing an oncogene/tumor-suppressor role in certain cancer types. For some of the other cancer types, however, a majority of oncogenes and tumor suppressors might be relevant. This is the case for rectum adenocarcinoma: its five oncogenes are BCL2, KIT, KLF4, MET, and PDGFRA. These genes are either linked to gastrointestinal cancer in the COSMIC database (BCL2, KIT and PDGFRA) or through literature findings (MET[68] and KLF4[69]).

Taking a closer look at the tumor suppressors, we found that at least two of these genes, CDKN2A[70] and SOCS1[64], have been linked to colorectal cancer. For the cancer types that performed the worst, liver hepatocellular carcinoma included none of the used oncogenes (AR, KLF4, PDGFRA, and RET) or tumor suppressors (BRCA2, CDKN2A, and TSC1) that were linked to it. This suggests that when a well-curated, cancer type specific list of oncogenes and tumor suppressors is present, Moonlight is successful in using gene expression data to detect the role of cancer driver genes. For rectum adenocarcinoma (one of the cancer types with the best performance), the top biological processes are able to cluster the two classes accurately (Fig. 6c).

**Integrating Connectivity Map to guide target therapies.** To capitalize on our discovery of dual-role cancer driver genes, we next employed Connectivity Map[71] to search for candidate compounds that could target cancer driver genes revealed by Moonlight (Methods). This tool provides a systematic approach for discovering associations among genes, chemicals, and

biological conditions. For the 776 biological mediators in breast cancer, this analysis revealed 365 compounds targeting 77 genes. We defined these 77 genes as critical drug genes, of which 18 were tumor suppressors and 59 oncogenes (Supplementary Data 12). Among the 365 compounds identified, 16 shared 26 mechanisms of action and targeted six tumor suppressors and 12 oncogenes (Fig. 7a, b). We observed that six compounds (methylnorlichexanthone, AG-879, axitinib, ENMD-2076, orantinib, and SU-1498) shared the VEGFR-inhibitor mechanism of action. Consequently, we speculate that a guided therapy of the mentioned drugs will be beneficial for breast-cancer treatment.

Furthermore, Connectivity Map also identified potential drugs to target the 151 dual-role genes identified by the expert-based Moonlight approach. For example, we identified ADRA2A, predicted as oncogene in breast cancer and tumor suppressor in bladder urothelial carcinoma, targeted by 62 compounds. In addition, PDGFRA was predicted to be oncogene in thyroid carcinoma and tumor suppressor in colon adenocarcinoma, targeted by 26 (Supplementary Data 12). Combining results from Moonlight and Connectivity Map potentially could help for drug-repurposing purposes.

**Cancer cell lines experiments validated cancer driver genes.** A major requirement for drug design is to functionally validate the inhibition potential of targeted cancer driver genes in ex vivo or in vivo cancer models.

Recently, multiple drugs were shown to act on the same cell lines in a first-of-its-kind study[72]. To aid in effective cancer treatments which concurrently activate tumor suppressors and inactivate oncogenes, novel drug-combination therapies are required. For this reason, we validated the predicted cancer driver genes in silico and further analyzed gene expression data from 1001 cancer cell lines retrieved from the Genomics of Drug Sensitivity in Cancer (GDSC) database[72]. We created a pipeline to automatically retrieve these data along with the gene expression matrix for 18 cancer types from GDSC data set (Methods).

Within these GDSC cell lines, we observed that 41% of the oncogenes upregulated in TCGA's breast-invasive carcinoma

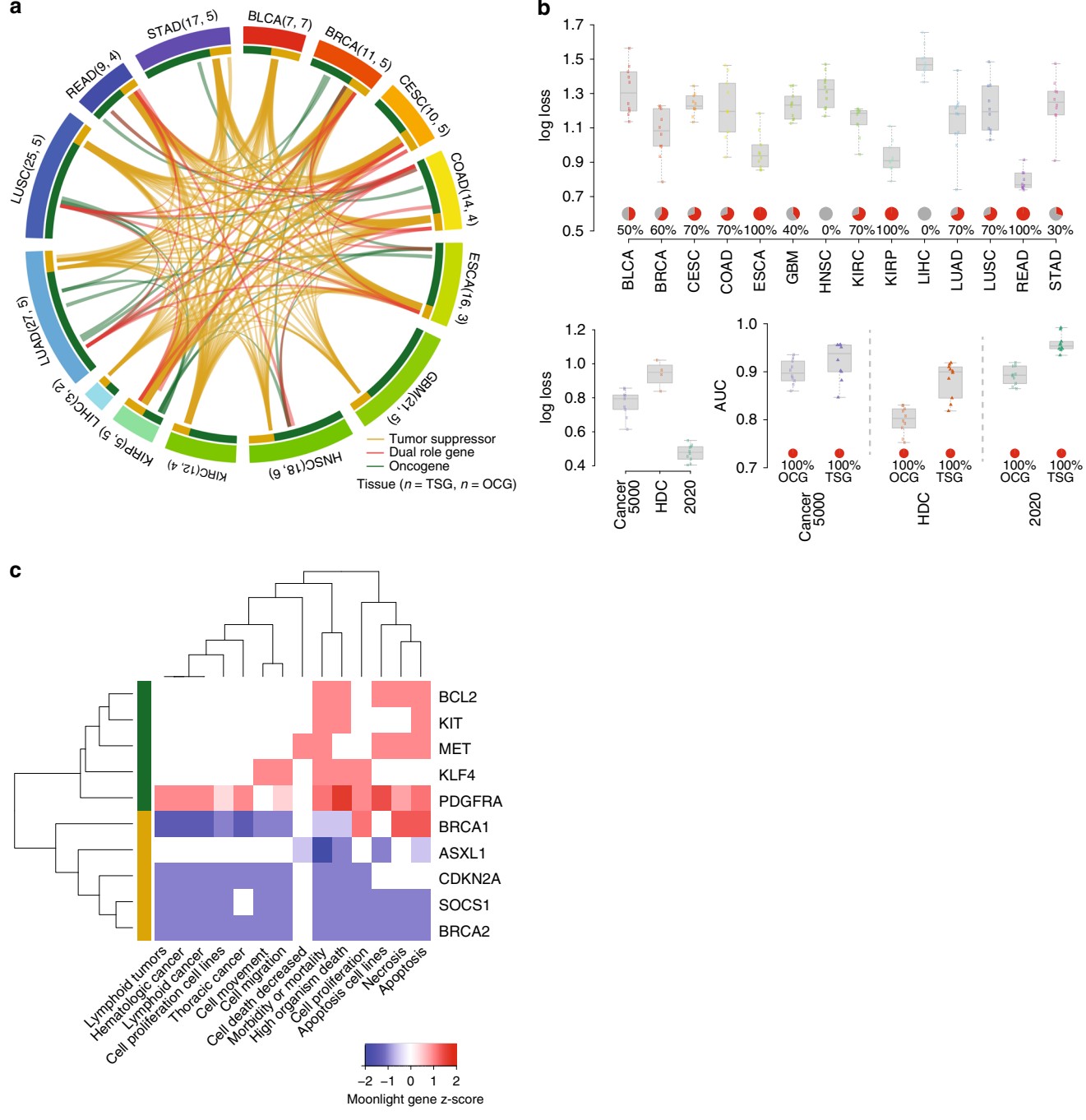

**Fig. 6 Moonlight a Pan-Cancer study: dual-role genes and machine-learning approach. a** Circos plot showing an integrative analysis of 14 TCGA cancer types using the ML approach. Labels around the plot specify the cancer type; the number of OCGs and TSGs for that cancer type are in parentheses. An edge is drawn in the center of the figure whenever the same gene is predicted in two different cancer types. Segments and edge colors correspond to cancer type: green (yellow) segments correspond to the number of OCGs (TSGs) predicted in that cancer type, and red edges represent dual-role genes. **b** Performance evaluation of Moonlight in terms of log loss for tumor suppressors and oncogenes predicted in 14 cancer types. Performance of 20/20 + and OncodriveRole in terms of log loss and AUC. **c** Heatmap showing Moonlight Gene Z-score for upstream regulators for rectum adenocarcinomas. Row colors indicate TSGs (yellow) and OCGs (green).

tumors had high expression. Simultaneously, 31% of the tumor suppressors downregulated had low expression (Methods; Fig. 2b, Fig. 7c; Supplementary Data 13). For example, Moonlight identified H2AF as a highly expressed oncogene in several breast-cancer cell lines. In contrast, Moonlight identified SOX7, CYP26B1, DACT2 as tumor suppressors with low expression in these same cell lines. These findings were also supported by literature [22,73–75].

## Discussion

In summary, Moonlight provides a platform for multi-omics integration and utilizes a wealth of prior knowledge (Fig. 1a). Such knowledge includes gene networks and ontologies unharnessed by many current bioinformatics tools for oncological discovery. Moonlight combines multiple functionalities to reproducibly integrate regulatory networks by means of gene expression, literature information, and evidence from multiple

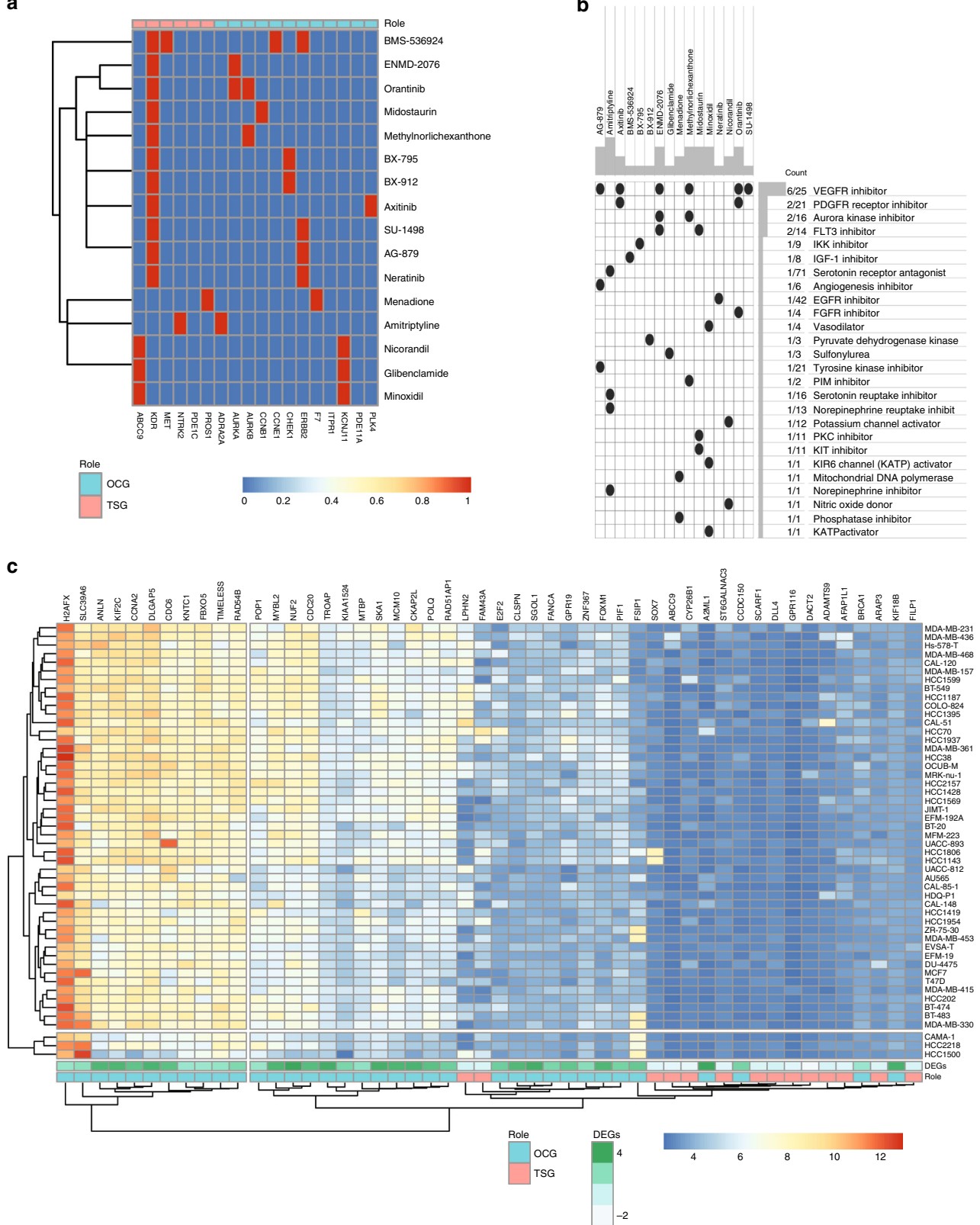

**Fig. 7 Moonlight intratumor heterogeneity, cell line, and drug analysis. a** Heatmap showing each compound (perturbagen) in rows from the Connectivity Map that share gene targets predicted as OCG (salmon) or TSG (teal) in columns. A red square indicates the presence of a relationship between compound and target. **b** Heatmap showing each compound (perturbagen) in columns from the Connectivity Map that shares mechanisms of action (rows), sorted by descending number of compounds with shared mechanisms of action. **c** Heatmap showing the top 50 TSG and OCG (by Moonlight Gene Z-score) predicted in breast cancer as mediators of apoptosis and proliferation (columns) and expression profiles of 50 breast-cancer cell lines from the Genomics of Drug Sensitivity in Cancer (GDSC) database (rows).

bulk-tumor -omics data (mutation, DNA methylation, chromatin accessibility, cell lines, and clinical data) (Methods; Fig. 1b). Because of Moonlight's ability to combine information from multiple sources, this software has the capability to define critical events when two or more key alterations appear.

Moonlight highlights cancer driver genes currently undetected by other tools and detects dual-role genes (oncogene in one cancer type and tumor suppressor in another). As a proof-of-principle, Moonlight accurately predicted cancer driver genes in breast-invasive carcinoma and 17 other cancer types, elucidating their underlying biological mechanisms. Moonlight successfully identified BCL2, SOX17, and ANGPTL4 as dual-role genes. These three genes show Moonlight's ability to detect complex interactions among biological process mediators, classifying oncogenes, and tumor suppressors. Analysis with Moonlight highlights the particular molecular changes associated to this dual-role effect. Proper evaluation of dual-role genes will allow for better comprehension of global tumor heterogeneity and will provide insights on tumor diagnosis, prognosis, and resistance to treatment ultimately leading to better therapeutic decisions.

In addition, we recently demonstrated the flexibility of Moonlight in pinpointing context-specific gene programs that are differentially expressed in varied scenarios from the TCGA Pan-Cancer Atlas Initiative. For instance, Moonlight extracted mutation-context differences in samples with and without mutations (somatic or germline) of BRCA1 and/or BRCA2, as well as in known cancer driver-gene mutations (e.g., missense or frameshift/nonsense)[10]. Also, Moonlight detected cell-of-origin differences based on stemness score associated with oncogenic dedifferentiation[76].

We further hypothesize that applying Moonlight to single-cell -omics data will reveal pathways and cancer driver genes that hide residual tumor cells and protect them from eradication by surgery, radiation, or chemotherapy. Another potential application of Moonlight is to gauge the impact of dual-role genes on tumor samples after polypharmacological treatments, as motivated by recent research[76]. Moreover, this information enables oncologists to choose the best personalized therapeutic option for each patient. Indeed, a therapy that has a positive effect on a subject could be completely inefficient on another tumor type due to the opposite behavior of the target protein. Apart from the inefficacy of the anticancer treatment, the use of off-targeted therapeutic options could have severe clinical consequences, such as toxicity or adverse side effects.

Even more critically, the existence of different cancer subtypes may affect patterns of mutations associated with drug resistance in rare cases. In addition, it has been reported that mutation of different amino acid sites are related to antibiotic drug resistance[77]. Interestingly, Moonlight identified GATA3 with three different mutation sites and predicted it correctly as an oncogene in breast cancer. Therefore, we speculate that designing specific drugs which target multiple amino acids enable more "stable" gene inactivation during therapy, and can overcome cancer-related drug resistance.

In addition, regulation of higher-order chromatin structures by DNA methylation and histone modification is crucial for genome reprogramming. Moonlight identified hypermethylated tumor suppressors and hypomethylated oncogenes. Interestingly, Moonlight detected open chromatin peaks in the intron regions for tumor suppressors. Also, Moonlight identified more mutations in intron regions than in promoter regions for tumor-suppressor genes. It is known that intron retention is a widespread mechanism of tumor-suppressor inactivation[78], which was consistent with our observation. This suggests that further investigation in long-range regulation within the intron region of tumor suppressors can inform us of the mechanism to re-activate silent tumor-suppressor genes.

When we explored the epigenetic modifiers or chromatin accessibility, we observed a global opening of chromatin in the promoter regions for oncogenes predicted by Moonlight. Concurrently, chromatin was more closed or had dampened signal for tumor suppressors. These findings confirmed the hypothesis that (i) a mechanism of activation for oncogenes is related to open chromatin in the promoter region, and (ii) distant chromatin peaks and open chromatin in intron regions are associated with tumor suppressors[79]. Therefore, our findings support that differential chromatin accessibility is an underlying biological mechanism of tumor suppressors and oncogenes. Recently, a Pan-Cancer analysis of 410 tumor samples in 23 cancer types showed that MYC, a well-known oncogene, had broad open chromatin in the promoter region[80]. Moonlight results support this finding.

Interestingly, a study has probed if it is possible for an oncogene to switch to a tumor suppressor[81]. The study showed that the epigenetic background of the cell type may only permit certain oncogenes or tumor suppressors to change roles. This perspective also applies to subtypes within a cancer type. For instance, some mutations are permissive in one subtype, whereas other alterations only work in other subtype. Their multiple findings agreed with Moonlight's findings, highlighting multiple genes identified as cancer driver genes (e.g., GATA3, CDH1, BRCA1, ESR1 in breast cancer[81]) that Moonlight predicted to drive tumorigenesis in breast and other cancer types.

As we look to the future of driver-gene discovery in cancer, tools like Moonlight will become essential for the integration of biological processes across many data molecular substrates. While our findings remain to be functionally validated, our tool has provided insights into genes that modulate proliferation, apoptosis, migration, and invasiveness. This hypothesis-generating mechanism provides clues to which gene properties that can be confirmed using in vivo models such as patient-derived tumors xenografted in mice, or proliferation assays in cell culture. Guided by Moonlight's in silico approach, functional studies will be more successful in identifying and confirming cancer biomarkers.

## Methods

**Moonlight workflow.** Here we describe the two Moonlight approaches: Moonlight-EB (expert based) and Moonlight-ML (machine learning) (Fig. 1b).

The EB and ML approaches share the following three initial steps (Fig. 1b; Methods): (i) Moonlight identifies a set of Differentially Expressed Genes (DEGs) between two conditions, then (ii) the gene expression data are used to infer a Gene Regulatory Network (GRN) with the DEGs as vertices, and (iii) using Functional Enrichment Analysis (FEA), Moonlight considers a DEG in a biological system and quantifies the DEG-BP (biological process) association with a Moonlight Process Z-score. Finally, we input DEGs and their GRN to Upstream Regulatory Analysis (URA), yielding upstream regulators of BPs mediated by the DEG and its targets.

The second part of the pipeline's tool provides pattern recognition analysis (PRA) that incorporates two approaches. In the first approach, PRA takes in two objects: (i) URA's output, and (ii) selection of a subset of the BP provided by the end user. In contrast, if the BPs are not provided, their selection is automated by an ML method (e.g., random forest model) trained on gold-standard oncogenes (OCG) and tumor-suppressor genes (TSG) in the second approach. In addition, dynamic recognition analysis (DRA) detects multiple patterns of BPs when different conditions are selected (Fig. 1b; Methods).

**State-of-the-art methods for cancer gene prediction.** Recent studies of tools predicting cancer driver genes using mutation, gene expression, and copy-number data are reported[66,82–84]. Table 1 shows a brief comparison of main current tools. These methods cover different methodological approaches: mutation-level threshold, mutation functional impact, and mutation and gene expression influence.

Among the state-of-the-art methods to identify cancer driver genes (CDGs), three of them have predicted the role of a CDG, such as TSG or OCG including 20/20[2], 20/20+[66], and OncodriveRole[67]. While these approaches are able to identify well-known cancer genes, they have difficulties when it comes to the prediction of new TSG/OCG candidates[85].

**Table 1 Comparison of tools used to predict cancer driver genes.**

| Method | Data type | Description |
|---|---|---|
| 20/20 | Mutation data | ≥20% truncating mutations is TSG; >20% missense mutations in recurrent positions is OCG |
| Oncodrive Role | Mutation and copy-number alteration data | Machine-learning approach using 30 features related to the pattern of alterations across tumors |
| ActiveDriver | Mutation data | Detecting cancer drivers based on unexpected mutation sites in phosphorylation regions |
| e-Driver | Mutation data | Identification of proteins with somatic missense mutations using domain based mutation analysis |
| MutSig2CV | Mutation and gene expression data | Identification of significantly mutated genes incorporating expression levels and replication times of DNA |
| DriverNet | Mutation, copy-number alteration, and gene expression data | Method that use interaction networks to identify mutated genes associated with the gene expression alterations of its known interacting genes |

The "20/20 rule" was proposed by Vogelstein et al.[2] to identify TSGs and OCGs based on their mutational pattern across tumor samples. If a gene has ≥20% truncating mutations, it is considered to be a TSG, whereas those with >20% missense mutations in recurrent positions are considered to be an OCG.

Schroeder et al. implemented OncodriveRole[67] to identify 30 features capable of differentiating between TSGs and OCGs. Successively, Tokheim et al. extended the original 20/20 rule[2] in an ML approach allowing the integration of multiple ratiometric features of positive selection in 20/20+[66] to predict oncogenes and TSGs from small somatic variants. The features capture mutational clustering, conservation, mutation in silico pathogenicity scores, mutation consequence types, protein interaction network connectivity, and other covariates (e.g., replication timing).

ActiveDriver and e-Driver identify driver genes detecting genes with mutations that might also have an impact on protein function. ActiveDriver detects driver genes with significantly higher mutation rates in posttranslationally modified sites such as phosphorylation-specific regions. e-Driver identifies protein regions (domains and disordered sites) enriched with somatic modifications that could influence protein function.

MutSig2CV and DriverNet detect driver genes integrating genomic and transcriptome data.

Compared with existing tools, Moonlight is able to extract, for each driver gene, the multilayer profile elucidating the BPs underlying their specific roles and interactions. Furthermore, the majority of the current methods use only mutation data to detect cancer drivers, limiting the knowledge of the related molecular mechanisms. Indeed, mutations can cause different effects such as a loss or reduction of mRNA transcripts impacting on the protein function. In line with this scenario to increase functional information and generate new hypotheses of gene function, transcriptome data have been used.

**Data sets and preprocessing.** The legacy level-3 data of the Pan-Cancer studies (18 cancer types), for which there were at least five samples of primary solid tumor (TP) or solid tissue normal (NT) available, were used in this study and downloaded in May 2018 from The Cancer Genome Atlas (TCGA) cohort deposited in the Genomic Data Commons (GDC) Data Portal (Supplementary Data 4).

RNA-seq raw counts of 7962 cases (7240 TP and 722 NT samples) aligned to the hg19 reference genome were downloaded from GDC's legacy archive, normalized, and filtered using the R/Bioconductor package TCGAbiolinks[14] version 2.9.5 using GDCquery(), GDCdownload(), and GDCprepare() functions for tumor types (level 3, and platform "IlluminaHiSeq_RNASeqV2"), as well as using data.type as "Gene expression quantification" and file.type as "results". This allowed for the extraction of the raw expression signal for expression of a gene for each case following the TCGA pipeline used to create level-3 expression data from RNA Sequence data. This pipeline used MapSplice[86] to do the alignment and RSEM to perform the quantification[87].

DNA methylation beta values of primary solid tumors (TP) and solid tissue normal (NT) from Pan-Cancer studies (18 cancer types) aligned to the hg19 reference genome were downloaded from GDC's legacy archive using the R/Bioconductor package TCGAbiolinks[14] version 2.9.5 using GDCquery(), GDCdownload(), and GDCprepare() functions for tumor types (level 3, and platform "Illumina Human Methylation 450"). This allowed for the extraction of the DNA methylation level-3 data following the TCGA pipeline used to create data from the Illumina Infinium HumanMethylation450 (HM450) array. This pipeline measured the level of methylation at known CpG sites as beta values, calculated from array intensities (level 2 data) as Beta = M/(M + U). Using probe sequence information provided in the manufacturer's manifest, HM450 probes were remapped to the hg19 reference genome[88]. Preprocessing steps included background correction, dye-bias normalization, and calculation of beta values. We used level-3 data. Beta values range from zero to one, with zero indicating no DNA methylation and one indicating complete DNA methylation.

Integrative analysis using mutation, clinical, and gene expression were performed following our recent TCGA's workflow[15].

For the intra-tumoral genomic and transcriptomic heterogeneity case study, we used Breast invasive carcinoma (BRCA) from TCGA as deposited in the GDC Data Portal. In particular, we downloaded, normalized, and filtered RNA-seq raw counts of 1211 BRCA cases as a legacy archive, using the reference of hg19, using the R/Bioconductor package TCGAbiolinks following the above pipeline. Among BRCA samples, 1097 were TP and 114 NT. The aggregation of the two matrices (tumor and normal) for both tumor types was then normalized using within-lane normalization to adjust for GC-content effect on read counts and upper-quantile between-lane normalization for distributional differences between lanes by applying the TCGAanalyze_Normalization() function adopting the EDASeq protocol[89,90]. Molecular subtypes, mutation data, and clinical data were extracted using TCGAbiolinks and the following functions: TCGAquery_subtype(), GDCquery_maf() (for retrieving somatic variants that were called by the MuTect2 pipeline), and GDCquery_clinic(), respectively. BRCA tumors with PAM50 classification[23] were stratified into five molecular subtypes: Basal-like (192), HER2-enriched (82), Luminal A (562), Luminal B (209), and Normal-like (40). We performed a comparison of each molecular subtype with normal samples excluding Normal-like subtypes.

**Biological processes.** To understand the molecular mechanisms that underlie CDGs, we focused our analysis on a subset of specific BPs. We used the function TCGAanalyze_DEA from TCGAbiolinks to create a merged list of all DEGs. Genes were identified as significantly differentially expressed if |logFC| ≥ 1 and FDR < 0.01 in at least one tumor type of the 18 different tumor types, which yielded 13,182 unique genes in total. We ran ingenuity pathway analysis (IPA)[91] for the above 13 k DEGs, which identified >500 relevant BPs in total (Supplementary Data 1). We then manually selected 101 BPs known to be relevant in cancer. A complete list of the chosen BPs is reported in Supplementary Data 2. For each BP, we provided the information whether its activation lead to cancer promotion or reduction according to current knowledge. For each gene/BP combination, we used IPA[91] to obtain the number of times (number of publications in PubMed) the pair was mentioned together in terms of upregulated, downregulated, or (less specifically) affected expression. We then employed Beegle[92] to allow the end user to update the mentioned number of times for BP.

**Gene programs.** To further investigate gene programs enriched by genes differentially expressed between two conditions, we employed Gene Set Enrichment Analysis (GSEA) for ten collections from the Molecular Signatures Database[93] as follows: H: hallmark gene sets, C2: BIOCARTA pathway database, C2: KEGG pathway database, C2: REACTOME pathway database, C3 TFT: transcription factor targets, C5 BP: GO BP, C5 CC: GO cellular component, C5 MF: GO molecular function, C6: oncogenic signatures, C7: immunologic signatures.

**Gold-standard gene set of driver genes.** A recent review[66] has argued that a comparative assessment of role prediction methods is not straightforward due to the lack of a clear gold standard of known OCGs and TSGs. To create the best currently available training set of known OCGs and TSGs, we used those genes in our training set that have been verified by at least two sources. We retrieved a first list of validated OCGs and TSGs from the Catalogue of Somatic Mutations in Cancer (COSMIC). The list consisted of 84 OCGs, 55 TSGs, 17 dual-role genes, and 439 genes without validated roles. The list provided additional information such as the type of mutation, either dominant (448), recessive (134), dominant/recessive (7), or undeclared (3). We downloaded a second list from Vogelstein et al.[2], where 54 OCGs and 71 TSGs were validated and recorded.

**Feature data from state-of-art cancer driver classification.** We downloaded the corresponding feature information from the supplementary material

**Table 2 Summary of TCGA RNA-seq samples and differentially expressed genes (DEGs), (tumor vs normal analysis) in 18 cancer types.**

| TCGA cancer type | Primary solid tumor (TP) | Solid tissue normal (NT) | DEG |
|---|---|---|---|
| BLCA | 408 | 19 | 2937 |
| BRCA | 1097 | 114 | 3390 |
| CHOL | 36 | 9 | 5015 |
| COAD | 286 | 41 | 3788 |
| ESCA | 184 | 11 | 2525 |
| GBM | 156 | 5 | 4828 |
| HNSC | 520 | 44 | 2973 |
| KICH | 66 | 25 | 4355 |
| KIRC | 533 | 72 | 3618 |
| KIRP | 290 | 32 | 3748 |
| LIHC | 371 | 50 | 3043 |
| LUAD | 515 | 59 | 3498 |
| LUSC | 503 | 51 | 4984 |
| PRAD | 497 | 52 | 1860 |
| READ | 94 | 10 | 3628 |
| STAD | 415 | 35 | 2622 |
| THCA | 505 | 59 | 1994 |
| UCEC | 176 | 24 | 4183 |

(http://karchinlab.org/data/Protocol/pancan-mutation-set-from-Tokheim-2016.txt.gz)[66]. This data set consists of 18,355 genes and 24 features which describe the mutations (defined in the original 20/20 rule paper[2]), gene length, gene degree, and betweenness based on information available from Biogrid[94] and the mean gene expression based on Cancer Cell Line Encyclopedia[95].

**Differential phenotypes analysis (DPA).** This function carries out two differential phenotypes analysis: if dataType is selected as "Gene Expression", it detects DEGs wrapping the function TCGAanalyze_DEA() from TCGAbiolinks. If dataType is selected as "Methylation", it detects differentially methylated regions (DMRs) wrapping the function TCGAanalyze_DMR() from TCGAbiolinks. The values generated from the differential expression analysis (DEA) analysis were sorted in ascending order and corrected using the Benjamini–Hochberg (BH) procedure for multiple-testing correction. We considered DEGs significant if the log fold change $|logFC| > 1$ and FDR < 0.01. The number of DEGs by cancer type for both OCG/TSG lists is presented in the first column of Table 2.

To identify DMRs, we used the Wilcoxon test followed by multiple testing using the BH method to estimate the false discovery rate. The default parameters for DMRs and methylated CpG sites, which are regarded as possible functional regions involved in gene transcriptional regulation, require a minimum absolute beta values delta of 0.2 and a false discovery rate (FDR)-adjusted Wilcoxon rank-sum $p < 0.01$ for the difference.

**Gene regulatory network (GRN).** We calculated the pairwise mutual information between the DEGs and all the genes filtered for each cancer type, considering only tumor samples. The pairwise mutual information was computed using entropy estimates from k-nearest neighbor distances ($k = 3$) with the R-package Parmigene[96] using the function GRN from MoonlightR. Afterwards, DEGs' regulon, representing the genes regulated by a DEG, are defined by filtering out non-significant (permutation $p > 0.05$) interactions using a permutation test (nboot = 100, nGenesPerm = 1000) and thus obtaining a set of regulated genes for each DEG.

**Functional enrichment analysis (FEA).** This analysis, using Fisher's test, allows for the identification of gene sets (with biological functions linked to cancer studies) that are significantly enriched in the regulated genes. The steps of FEA involve (i) evaluating if DEGs are involved in a BP through an assessment of the overlap between the list of DEGs and genes relevant to this BP determined by literature mining, and (ii) detecting the BPs mainly enriched by DEGs. A Fisher exact test is used to calculate the probability of the BP's enrichment based on the overlapping of the genes annotated in each BP and the entire list of DEGs. We considered BPs enriched significantly with $|Moonlight-score| >= 1$ and FDR $<= 0.01$.

**Upstream regulator analysis (URA).** This analysis is carried out for each differentially expressed gene $i$ and each BP $j$. As a first step, genes in the network that are connected to gene $i$ are selected and form $\mathbf{S}_i$. We then carry out a functional enrichment analysis computing a Moonlight Process Z-score that compares the literature-based knowledge to the result of the differential expression analysis.

Let $L_{kj}$ be the result of the IPA-based literature mining for gene $k$ and BP $j$: $L_{kj} \in$ {increased, decreased, affected}. Let

$$y_{kj} = 1 \text{ if } (L_{kj} = \text{increased } \& \log FC(k) > 0)$$
$$\text{or } (L_{kj} = \text{decreased } \& \log FC(k) < 0), \tag{1a}$$

$$y_{kj} = -1 \text{ if} (L_{kj} = \text{increased } \& \log FC(k) < 0)$$
$$\text{or } (L_{kj} = \text{decreased } \& \log FC(k) > 0), \tag{1b}$$

$$y_{kj} = 0 \text{ if} (L_{kj} = \text{affected } \& \log FC(k) = 0) \cdot \tag{1c}$$

Let $n$ be the number of genes in $\mathbf{S}_i$ for which the literature mining has support for either "Decreased" or "Increased" effect in the process $BP_j$. The Moonlight Gene Z-score for each gene $i$ to BP $j$ pair is computed as

$$\text{Moonlight Gene Z-score}_{ij} = \frac{\sum_{k \in S_i} y_{kj}}{\sqrt{n}}. \tag{2}$$

**Literature phenotype analysis (LPA).** As described in the Biological Processes section, we extracted 101 BPs (reported in Supplementary Data 5) using IPA[91] that were successively used for the downstream analysis. LPA interrogates PubMed to obtain a table with information for each gene and a particular BP such as apoptosis or proliferation to understand the number of publications reporting the relationship of a gene-BP (increasing, decreasing, or affected). To filter out false positives obtained from text co-occurrence, it is possible to integrate Beegle's[92] results applied on individual BP, considering the overlapping results. Here with the LPA function, it is possible to extract a BP-genes database from the literature with a twofold aim: (i) producing updated literature information, and (ii) flexibility for BPs of relevant interest.

**Pattern recognition analysis (PRA).** PRA allows for the identification of a list of TSGs and OCGs when BPs are provided such as apoptosis and proliferation, otherwise a random Forest-based classifier can be used on new data. We define a pattern when a group of genes classified as OCGs share similar BP as apoptosis (DOWN) and proliferation (UP) while genes classified as TSGs share apoptosis (UP) and proliferation (DOWN).

**Dynamic recognition analysis (DRA).** This analysis detects multiple patterns of BPs when different conditions are selected. For the breast-cancer molecular subtypes application, we used fgsea package with the ten collections from the Molecular Signatures Database[93] using the following parameters: minSize = 15, maxSize = 500, and nperm=1000. Categories were considered significantly enriched with permutation $P < 0.05$.

**ROMA score for pathway activity.** For the pathway activity evaluation, Representation and quantification Of Module Activity (ROMA) (https://github.com/sysbio-curie/Roma)[97] was also employed as an alternative to the Moonlight Process Z-score. For each module under analysis, the algorithm applies principal component analysis to the sub-matrix composed of the expression values of the signature genes across samples. ROMA then evaluates the module overdispersion by verifying if the amount of variance explained by the first principal component of the expression sub-matrix (L1 value in ROMA) is significantly larger than that of a random set of genes of the same size. This represents an unsupervised approach that can be used in combination with the supervised Moonlight Process Z-score to detect concordant signals. An example of application of ROMA to TCGA breast-invasive carcinoma is shown in Supplementary Data 14, where the ROMA activity score of biologically processes potentially modulated by cancer driver genes is reported.

**Machine-learning approach.** We used the Moonlight Process Z-score matrix as input to the random forest procedure, such that the BPs are the features that the learning method can include in the model. The obtained model can then be used to predict the role of genes that were not included in model building and obtain a relevance score for each of the BPs. This model is trained on a gold-standard gene set of known OCGs and TSGs based on the intersection of two sources: (i) the list provided by the COSMIC database[98,99], and (ii) the cancer genes identified by Vogelstein et al.[2].

**Evaluation criteria.** We used two different quality measures in our evaluation. The first one is the multi-class log-loss measure. The lower the log-loss value, the better the model's performance. The log loss is defined as:

$$-\frac{1}{m} \sum_{i=1}^{m} \sum_{j=1}^{3} y_{ij} log(p_{ij}), \tag{3}$$

where $y_{ij}$ is a binary variable, that is equal to one when $i = j$ and zero otherwise. The probability of gene $i$ to be in class $j$ is denoted by $p_{ij}$. For each gene, we compute the logarithm of the probability that gene $i$ belongs to class $j$ according to

our prediction. We then sum over all classes (three in our case), adding the log value to the log loss if gene $i$ belongs to class $j$ according to the known truth. Then we average over all genes ($m$) and finally take the negative value of the obtained score.

The logarithm of a high value is considerably lower than the logarithm of a low probability ($\log(1) = 0$, $\log(x) \to -\infty$ as $x \to +0$). Therefore, when the prediction of the model strongly disagrees with the actual class, the impact on the log-loss measure will be high. This measure penalizes strongly confident misclassifications. The second measure we use is the area under the ROC curve in a one-versus-all strategy. We are most interested in the performance of OCGs and TSGs and thus evaluated the total score as an average over these two classes.

Lastly, we compare the obtained results in each run with a set of random classifications. We generate the random predictions by randomly assigning gene names to the data that is used to train the random forest model. We repeat this procedure 100 times for each of the ten repetitions. The estimate of $p$ for log-loss evaluation is obtained by computing

$$\frac{\sum_{i=1}^{100} \mathrm{I}\{\mathrm{res}_{\mathrm{loo}} \geq \mathrm{res}_{\mathrm{loorandom}_i}\}}{100}. \quad (4)$$

The estimate of $p$ for the AUC evaluation is obtained computing

$$\frac{\sum_{i=1}^{100} \mathrm{I}\{\mathrm{res}_{\mathrm{loo}} \leq \mathrm{res}_{\mathrm{loorandom}_i}\}}{100}. \quad (5)$$

**Moonlight's performance**. To evaluate Moonlight's performance, we applied the same ML approach we used for Moonlight to the data used by 20/20+[66], and OncodriveRole[67] carrying out a leave-one-out cross-validation scheme. We repeated the procedure 10 times, each time undersampling the two majority classes (OCGs and neutral genes). We assessed the results using two different quality measures, i.e., log loss and AUC (one class versus all). Furthermore, we compared the results to randomized Moonlight Gene Z-score matrices and to the state-of-the-art methods 20/20+[66] and OncodriveRole[67]. Finally, we used the complete training data to predict dual-role genes in different cancer types and compare the obtained genes to those dual genes already known in the literature.

**Mutation analysis**. We integrated a publicly available MAF file (syn7824274, https://gdc.cancer.gov/about-data/publications/mc3-2017) that was recently compiled by the TCGA MC3 Working Group and is annotated with filter flags to highlight potential artifacts or discrepancies. This data set represents the most uniform attempt to systematically provide mutation calls for TCGA tumors. The MC3 effort provided consensus calls of variants from seven software packages: MuTect, MuSE, VarScan2, Radia, Pindel, Somatic Sniper, and Indelocator[100].

We then integrated cancer driver genes, predicted by Moonlight using RNA-seq's data. Boxplot was generated using the function ggplot from the ggplot2 package and the function ggpubr. P-values were generated using the function stat_compare_means from ggpubr with $t$ test method to compare means.

**Copy-number analysis**. We used TCGAbiolinks to retrieve the performed CNA analysis using gene level CNA results from GISTIC2.0[101] for the 18 cancer types and the function TCGAvisualize_CN to plot the amplified (top) and deleted genes (bottom). The genome is oriented horizontally from top to the bottom, and GISTIC $q$-values at each locus are plotted from the left to right on a log scale. The orange line represents the significance threshold ($q$-value = 0.25). We annotated the gene in the broad peak using the function findOverlaps from the package GenomicRanges.

**Chromatin accessibility analysis**. We used TCGAbiolinks to retrieve and analyze the ATAC-seq bigWig track files for all the TCGA Pan-Cancer types available. Genome browser screenshots of normalized ATAC-seq sequencing tracks of ten different breast-cancer samples, shown across the same genes locus, were generated using UCSC Genome Browser v.376[102]. We used the function TCGAquery_subtype from TCGAbiolinks to stratify the BRCA samples in molecular-subtype samples according to the PAM50 classification and we classified the basal samples according the Triple-Negative Breast Cancer Lehmann's subtypes[103] using the tool TNBCtype[104]. Color code is according to TCGA BRCA molecular subtypes.

**Survival analysis**. We used TCGAbiolinks with the clinical data to analyze the survival curves for the 33% of patients with higher expression of a specific gene versus the 33% with lower expression using the function TCGAanalyze_divideGroups(). The associations between higher and lower expression of a specific gene, if predicted as OCG or TCG, in primary tumors were evaluated in Pan-Cancer data with the function TCGAanalyze_SurvivalKM(). Kaplan–Meier plots showing the association of a specific gene expression and other clinical parameters with patient survival were performed using the function TCGAanalyze_survival() reporting the log-rank test $p$s. If a CDG had a log-rank test $p < 0.05$ and high expression was related to better outcome, we reported it in the table as a good prognosis. If a CDG had a log-rank test $p < 0.05$ and high expression was related to worse outcome, we reported it in the table as a poor prognosis.

**Cell-line analysis**. RMA normalized expression data for 1001 Cell lines from the Genomics of Drug Sensitivity in Cancer's study[72], was downloaded from ftp://ftp.sanger.ac.uk/pub/project/cancerrxgene/releases/current_release/sanger1018_brainarray_ensemblgene_rma.txt.gz. Annotation of cell lines were considered with TCGA's classification as reported in ftp://ftp.sanger.ac.uk/pub/project/cancerrxgene/releases/current_release/Cell_Lines_Details.xlsx. Genes with a mean expression of less than 25% of the quantile expression distribution were considered lowly expressed in cell lines while genes with a mean expression of more than 75% were considered highly expressed.

**Connectivity MAP analysis**. We used the Broad Institute's Connectivity Map build 02[105], a public online tool (https://portals.broadinstitute.org/cmap/) (with registration) that allows users to predict compounds that can activate or inhibit cancer driver genes based on a gene expression signature. To further investigate the mechanism of actions and drug targets, we performed specific analysis within Connectivity Map tools (https://clue.io/)[71].

**Reporting summary**. Further information on research design is available in the Nature Research Reporting Summary linked to this article.

## Data availability

The -omics data sets (gene expression, methylation, copy number, chromatin accessibility, clinical, and mutation) analyzed during this study are publicly available in the repository https://portal.gdc.cancer.gov/ and can be downloaded directly by using the TCGAbiolinks R package as described in the Methods section. The cell lines data set analyzed during this study are publicly available in the repository https://www.cancerrxgene.org/downloads. All data generated or analyzed during this study are included in this published article, its supplementary information files, and in the publication folder https://github.com/ibsquare/.

## Code availability

Updated links to the packages and tutorials related to Moonlight are available within the Bioconductor project at http://bioconductor.org/packages/MoonlightR/ and in GitHub https://github.com/ibsquare/MoonlightR. The package vignette with R scripts to reproduce the results and figures at the time of publication are provided as Supplementary. Data with intermediate results and code to generate specific analysis are available from the corresponding author, Dr. Antonio Colaprico, and will be uploaded to GitHub [https://github.com/torongs82/] upon request.

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

## Acknowledgements

We are grateful to Matthieu Defrance, Kridsadakorn Chaichoompu, Kristel Van Steen, Benjamin Haibe-Kains ans Thuc Duy Le for suggestions and scientific advice in the Moonlight project. We would also like to thank Lisa Cantwell for her scientific proof-reading of the paper. The project was supported by the BridgeIRIS project [http://mlg.ulb.ac.be/BridgeIRIS], funded by INNOVIRIS, Region de Bruxelles Capitale, Brussels, Belgium, and by GENGISCAN: GENomic profiling of Gastrointestinal Inflammatory-Sensitive CANcers, [http://mlg.ulb.ac.be/GENGISCAN] Belgian FNRS PDR (T100914F to A.C., C.O., and Gi.B.). Gi.B. was also supported by the project WALINNOV 2017 – N° 1710030 - CAUSEL I.C, C.C. and Gl.B. were supported by INTEROMICS flagship project (http://www.interomics.eu/it/home), National Research Council CUP Grant B91J12000190001, and the project grant SysBioNet, Italian Roadmap Research Infra-structures 2012. A.C., G.O., and X.C. were supported by grants from NCI R01CA200987, R01CA158472, and U24CA210954. E.P.'s group is supported by grants from LEO Foundation (grant number LF17006), the Innovation Fund Denmark (grant number 5189-00052B), and the Danish National Research Foundation (DNRF125).

## Author contributions

A.C. envisioned Moonlight, conceived the project, and performed chromatin accessibility, DNA methylation, copy-number variation, cell line, survival and drug analysis. A.C. and C.O. developed the method and designed the experiments. A.C., C.O, C.C, T.T., T.C.S., A.V.O., and L.C. performed computational analysis using gene expression data and implemented the software tool as R/Bioconductor package. L.C. performed ROMA analysis. A.C., C.O., C.C. and Gl.B. designed and performed research and interpreted the data results. C.C. and Gl.B curated the BPs data sets and scored the data. C.O. assessed the performance and accuracy of the method. A.C., C.O. T.C.S., and M.H.B. assembled the display (figures and tables) items. A.C., C.O., M.H.B., T.C.S., G.J.O., and E.P. wrote the paper with input from all other authors. Gi.B, X.C.,. and E.P. supervised the study. Gi.B., E.P., X.C., G.J.O., H.N, I.C., Gi.B., E.B., and A.Z. provided scientific and technical advice. All authors read and approved the final paper.

## Competing interests

The authors declare no competing interests.
