## [Peer Review File · Nature Communications]

Reviewers' comments:

Reviewer #2 (Remarks to the Author):

The authors developed a tool (called Moonlight) by comprehensively considering multiple -omics data to discover cancer driver genes (CDGs). Especially, by view of some prior information related to cancer biological processes (BPs) they distinguished oncogenes (OCGs), tumor suppressor genes (TSGs) and dual-role genes. Such exploration would benefit to unveil and understand cancer mechanisms. However, such a tool may fit in a more specialized journal about cancer bioinformatics. More comments are listed as follows:

1. Generally, many computational methods and tools have been developed for discover cancer driver genes and pathways. The authors should well acknowledge those contributions carefully and clarify the novelty of this tool.
2. In Figure 1 on Page 28, the subfigure a) is not necessarily included here, because some kinds of data are actually not used in this manuscript, or it may be better to delete the unused data types to avoid misunderstanding here.
3. Some descriptions for the tool are not quite clear. For example, according to the Section Methods, for each gene i and BP j , Moonlight Z – score ij can be calculated on Page 18. In Figure 2a) on Page 29, I wonder how to get the Moonlight Z – score value. Is it the sum for all genes included in the corresponding BP?
4. The description and the meaning of the formula for the Evaluation criteria on Page 20 is not clear either.
5. In Section Biological Processes (BPs) on Page 15, 13182 differentially expressed genes were classified into two groups by $|\logFC| \geq 1.5$ (7368 DEGs) and $|\logFC| \leq 1.5$ (5814 genes). $|\logFC| \leq 1.5$ also corresponds to differentially expressed genes?
6. By applying moonlight, many predicted OCGs and TSGs are potentially involved in one BP regulation. For example, it is well known that apoptosis is often down regulated in cancers. Here in Figure 2b) on Page 29, how to judge the BP apoptosis is actually up regulated or down regulated in breast cancer?
7. On Page 7, it is known that hypermethylation is generally as an important epigenetic control mechanism for TSGs inactivation in cancer cells. Why this implies a possible mutation in the corresponding CDGs? The authors had better give some explanations.
8. The manuscript needs to be improved. There are some typos in the text.

Reviewer #3 (Remarks to the Author):

In this study, the authors analyzed genes coding for protein with possible dual-function as oncogenes in one cancer type and tumor suppressor genes in another cancer type. They developed

novel bioinformatic tool (Moonlight) and used it to identify the dual-function genes using RNA expression and DNA methylation datasets from TCGA. Interestingly, the authors identified 160 dual-role genes in 8,000 samples from 18 cancer types. Finally, they connected these genes with the biological mechanisms and survival of the cancer patients. The authors took advantage of the Connectivity Map and identified potential drug candidates that target the dual-role genes.

I find the underlying biological questions and the computational extraction of dual-role genes function from large tumor datasets of high interest and importance. However, the performed analyses and their justification are insufficient and not convincing. Furthermore, the manuscript mainly focuses on the ability to identify the dual-role genes rather than on learned biology and mechanisms of cancer progression.

Overall, I strongly believe this work is of high importance but require additional effort before publication in Nature Communications, in order to be appealing for broader biomedical audience.

Major comments:

1. The authors should add more biological interpretation to the presented data. Do not focus solely on the technical aspects, amount of data analyzed and generated. Provide with more examples of factors for which dual-role is has been already identified and described in literature.
2. The authors should assign biological functional classes to the identified dual-role genes; e.g. transcription factors, kinases/lipid kinases, epigenetic modifiers. Present the data in one clear table as figure. The supplementary figures are useful for data mining but not to follow the text.
3. Can miRNAs or lncRNAs that play role as dual-role factors in cancer progression? Could some examples be identified using Moonlight analyses?
4. If possible, identify molecular changes (mutations, deletions, amplifications, promoter DNA methylation) that are specific for the dual-role genes in comparison to single-role oncogenes. For example, is TSG function of dual-role genes modulated by promoter methylation or inactivating mutations, and/or deletions.
5. The discussion section requires more biomedical interpretation of the obtained results. Currently, there is too much focus on the descriptive abilities of the Moonlight rather than on the biological findings it delivers. The authors should highlight known dual-role genes that were identified by Moonlight that validate the methodology. On the other hand, authors should discuss more biological function of newly identified dual-role genes with potential clinical implications; e.g. these that correlate with the survival of cancer patients. Also, disease-free survival should be used for the statistical analysis, if possible.

Minor comments:

1. The abbreviations for gene types and processes are overused and should be limited when really needed. It will improve the flow of the text and will make it less technical.

NCOMMS-18-30511-T "Moonlight: a tool to interpret pathways to discover cancer driver genes"

To aid the reviewer in understanding our response, we use the following shorthand to reference subsections of the Results section:

1. Overview of Moonlight - R1
2. Application of Moonlight Strategy to Pan-Cancer and Breast Cancer Data - R2
3. Novel oncogenic mediators identified by Moonlight controlling cancer gene programs in breast cancer - R3
4. Identified cancer-driver genes are associated with breast cancer heterogeneity - R4
5. DNA methylation changes control activities of cancer-driver genes - R5
6. Cancer-driver genes are prioritized at distinct chromatin accessibility regions - R6
7. Critical cancer-driver genes reshape the copy-number landscape - R7
8. Oncogenic mediators exhibited differences in somatic and intron mutations - R8
9. Oncogenic mediators impaired survival outcomes - R9
10. Moonlight machine-learning approach and comparison with other tools - R10
11. Connectivity Map collaborated with Moonlight to inform targets for novel guided therapies along with potentially effective drug-repurposing - R11
12. Cancer cell lines experiments validated cancer-driver genes - R12

Reviewer 2

The authors developed a tool (called Moonlight) by comprehensively considering multiple -omics data to discover cancer driver genes (CDGs). Especially, by view of some prior information related to cancer biological processes (BPs) they distinguished oncogenes (OCGs), tumor suppressor genes (TSGs) and dual-role genes. Such exploration would benefit to unveil and understand cancer mechanisms. However, such a tool may fit in a more specialized journal about cancer bioinformatics. More comments are listed as follows:

2.1. Generally, many computational methods and tools have been developed for discover cancer driver genes and pathways. The authors should well acknowledge those contributions carefully and clarify the novelty of this tool.

We thank the reviewer for this comment. We agree that comparisons between various tools are highly informative. However, our aim for this manuscript was for discovery, as such, we did not intend to do an extensive method comparison, which has already been done by several recent publications (PMIDs: 27417679, 25348012, 28714987, 27911828). Withal to address this point, we have incorporated a more comprehensive tool description in the **Methods** section. In particular, we reported that several approaches have been developed to discover cancer-driver genes and pathways, but these methods have not considered how biological processes are affected by gene deregulation in cancer. The novelty of our approach is the interpretation of pathways to identify key cancer-driver genes while also integrating information from interacting genes and transcriptomic data (**R1**, **R3**). These methods cover different methodological approaches: mutation-level threshold, mutation functional impact, and mutation and gene expression influence. Compared to existing tools, Moonlight is able to extract the multi-layer profile for each driver gene by elucidating the biological processes underlying their specific roles and their interactions along with the novelty of identification of dual-role genes. Furthermore, the majority of the current methods use mutation data alone to detect cancer drivers, limiting the knowledge of the related molecular mechanisms. Indeed, mutations can cause different effects such as a loss of mRNA transcripts

impacting protein function. Thus, to increase functional information and generate new hypotheses of gene function, Moonlight uses transcriptome and other data sources.

2.2. In Figure 1 on Page 28, the subfigure a) is not necessarily included here, because some kinds of data are actually not used in this manuscript, or it may be better to delete the unused data types to avoid misunderstanding here.

At the reviewer's suggestion, we have updated **Figure 1a**, reflecting the data that Moonlight integrated to discover and characterize oncogenic mediators and critical cancer driver genes. In particular, we removed protein and microRNA data—as we did not use them in the current version of the article, and we included chromatin accessibility data. We have splitted Figure 1a in two parts (**Discovery and Mechanistic Indicators**) to clarify that the end-user needs only gene expression data to discover oncogenic mediators. This is because users can employ other data types as available (i.e., epigenetic changes, methylation, copy number, chromatin accessibility, etc.) to confirm the oncogenic mediators, namely mechanistic indicators.

2.3. Some descriptions for the tool are not quite clear. For example, according to the Section Methods, for each gene i and BP j , Moonlight Z – score ij can be calculated on Page 18. In Figure 2a) on Page 29, I wonder how to get the Moonlight Z – score value. Is it the sum for all genes included in the corresponding BP?

In light of this comment, we have updated the formula for Moonlight Z-score **in the Methods subsection (Upstream regulator analysis (URA))**.

In summary Moonlight Z-score can be considered as the formula below, that (i) take the sum of the positive correlation between the differentially expressed gene (logFC) annotated to the biological process and the information from literature, if that gene was increased the biological process 1, decreased -1, if no change 0, (ii) take the sum divided by the sqrt (of all the genes associated to the the biological process) We also reported in the **Methods** the description of the formula with more details.

This is the formula from our R function FEA from MoonlightR package:

```
(Zscore <- sum(selected_diseases$Correlation)/sqrt(length(PredictionIncreased) +  
length(PredictionDecreased)))
```

2.4. The description and the meaning of the formula for the Evaluation criteria on Page 20 is not clear either.

To address the reviewer's valid critique, we have updated the formula for the Evaluation criteria in the **Methods** subsection.

2.5. In Section Biological Processes (BPs) on Page 15, 13182 differentially expressed genes were classified into two groups by $|\logFC| \geq 1.5$ (7368 DEGs) and $|\logFC| \leq 1.5$ (5814 genes). $|\logFC| \leq 1.5$ also corresponds to differentially expressed genes?

The reviewer's question motivated us to clarify the differentially expressed genes used in the **Methods** subsection on Biological Processes. We used the function TCGAanalyze_DEA from TCGAbiolinks to create a merged list of all Differentially Expressed Genes (DEGs). Genes were identified as significantly differentially expressed if $|\logFC| \geq 1$ and FDR < 0.01 in at least one tumor type of the 18 different tumor

types, which yielded 13182 unique genes in total. We have also provided the full DEGs list for each tumor types in **Supplementary Table 3**.

2.6. By applying moonlight, many predicted OCGs and TSGs are potentially involved in one BP regulation. For example, it is well known that apoptosis is often down regulated in cancers. Here in Figure 2b) on Page 29, how to judge the BP apoptosis is actually up regulated or down regulated in breast cancer?

We thank the reviewer for this question. We have investigated in more detail apoptosis and proliferation modulated by Cancer Driver Genes (CDGs). It is well known that apoptosis, a key biological process, is often downregulated in cancer. To confirm that our approach identified important biological processes regulated by CDGs, we considered apoptosis. Looking more in depth to the functions of the genes identified by Moonlight analysis, among the 38 hypomethylated, upregulated oncogenes (**Figure 2b**), we found several anti-apoptotic genes, including CDC6 (PMID: 16801388), FOXM1 (PMID: 22393369), and others (**R3, R4**). In the same view, among the 12 hypermethylated, downregulated tumor suppressors (TSGs), we found a number of pro-apoptotic genes, including DACT2 (PMID: 28796248), SCARF1 (PMID: 23892722), and others (**R12, Discussion**). The upregulation of anti-apoptotic genes or the downregulation of pro-apoptotic genes could lead to the suppression of apoptosis in cancer, confirming the findings described in the literature. Based on these findings, we speculate that the final outcome of apoptotic inhibition is due to the overall balance between upregulation of anti-apoptotic oncogenes and downregulation of proapoptotic tumor suppressors.

2.7. On Page 7, it is known that hypermethylation is generally as an important epigenetic control mechanism for TSGs inactivation in cancer cells. Why this implies a possible mutation in the corresponding CDGs? The authors had better give some explanations.

The reviewer raises a valid concern. Motivated by the reviewer's comment, we have incorporated two novel sections about epigenetic changes and mutations (**R5, R8**). It is true that inactivation of TSG by hypermethylation does not imply a possible mutation in the TSG. For this reason, we have explored different mutation types for each oncogenic mediators, discovering that oncogenes were associated to more missense mutations whereas TSG were associated to more intron mutations. In future work, we aim to investigate specific DNA damage pathways and implicated cancer driver genes.

2.8. The manuscript needs to be improved. There are some typos in the text.

To answer the reviewer's critique, we have collaborated with native English speakers with scientific writing expertise to extensively re-write and edit the entire manuscript.

Reviewer 3

In this study, the authors analyzed genes coding for protein with possible dual-function as oncogenes in one cancer type and tumor suppressor genes in another cancer type. They developed novel bioinformatic tool (Moonlight) and used it to identify the dual-function genes using RNA expression and DNA methylation datasets from TCGA. Interestingly, the authors identified 160 dual-role genes in 8,000 samples from 18 cancer types. Finally, they connected these genes with the biological mechanisms and survival of the cancer patients. The authors took advantage of the Connectivity Map and identified potential drug candidates that target the dual-role genes.

I find the underlying biological questions and the computational extraction of dual-role genes function from large tumor datasets of high interest and importance. However, the performed analyses and their justification are insufficient and not convincing. Furthermore, the manuscript mainly focuses on the ability to identify the dual-role genes rather than on learned biology and mechanisms of cancer progression.

Overall, I strongly believe this work is of high importance but require additional effort before publication in Nature Communications, in order to be appealing for broader biomedical audience.

Major comments:

3.1. The authors should add more biological interpretation to the presented data. Do not focus solely on the technical aspects, amount of data analyzed and generated. Provide with more examples of factors for which dual-role is has been already identified and described in literature.

We thank the reviewer for this comment. We have incorporated a major revision of the results section, integrating chromatin accessibility (**R6**), copy number changes (**R7**), and mutation data (**R8**). We Include more biological interpretation and discussion of the presented data throughout the Results and Discussion sections.

Firstly, LSM1, a gene predicted by Moonlight as an oncogene and reported in literature as oncogene in breast cancer [PMID: 17001308], showed the highest peak in the promoter region by ATAC-seq chromatin accessibility. Additionally, we reported several examples of dual-role genes that Moonlight discovered that have already been identified and described in literature (eg. GAS7, ANGPTL4, SOX17, BCL2, CDKN2A, KIT, and SOCS1). Moonlight identified GAS7 as a hypermethylated tumor suppressor in lung cancer and as an hypomethylated oncogene in head and neck squamous cell tumors. These findings were confirmed literature [PMID: 26506240]. Also, Moonlight predicted ANGPTL4 to be oncogene in kidney cancers with associated promoter peak and tumor suppressor in prostate adenocarcinoma with hypermethylation in the promoter region. This was also confirmed by a recent study in literature [PMID: 28182091] A similar behavior was observed for SOX17, a transcriptional regulator, which Moonlight predicted as an oncogene in uterine corpus endometrial carcinoma associated with promoter peak and a tumor suppressor associated with hypermethylation in lung squamous cell carcinoma. These findings were confirmed by a recent study using ChipSeq of SOX17 in endometrial cancer,[PMID: 30356064] and lung cancer.[PMID: 22846201]. Finally, Moonlight showed that BCL2, a well-known mediator of apoptosis, is a dual-role gene. Specifically, Moonlight identified BLC2 as an oncogene in thyroid carcinoma, through decreasing apoptosis and showing a peak in the exon region concurrently. This finding was independently reported in literature [PMID: 8695923].

3.2. The authors should assign biological functional classes to the identified dual-role genes; e.g. transcription factors, kinases/lipid kinases, epigenetic modifiers. Present the data in one clear table as figure. The supplementary figures are useful for data mining but not to follow the text.

Inspired by this comment, we have incorporated the biological functional classes for each oncogenic mediator, cancer driver gene, and dual role gene identified by Moonlight [Supplementary Table 6-13]. We have presented the data and results in multiple supplementary tables including copy number changes, DNA methylation, chromatin accessibility, and others.

3.3. Can miRNAs or lncRNAs that play role as dual-role factors in cancer progression? Could some examples be identified using Moonlight analyses?

We thank the reviewer for this valuable question. We have identified one lncRNAs (**R7**) as dual role (ADAMTS4). We believe that multiple lncRNAs as well as miRNAs can be identified in a cross-talk regulation of the biological processes, as we found for genes. In this way, non-coding RNA as miRNAs can also be identified as oncogenic mediators by Moonlight. In future work, we wish to extend our study to understand and identify miRNAs or lncRNAs that can exhibit dual-role behavior in cancer progression. To accomplish this task, a comparison between each stage (I-IV), or grade (I-IV), versus normal tissue will be essential to identify specific miRNAs or lncRNAs differentially expressed within stages or grades. In addition, we will extend Moonlight with the comparison of specific molecular subtypes for each cancer study compared to normal tissue, for miRNA and lncRNA data. In this way, it will be possible to identify dual role miRNAs or lncRNAs that can act a dual-role in cancer progression.

3.4. If possible, identify molecular changes (mutations, deletions, amplifications, promoter DNA methylation) that are specific for the dual-role genes in comparison to single-role oncogenes. For example, is TSG function of dual-role genes modulated by promoter methylation or inactivating mutations, and/or deletions.

This comment motivated us to add multiple new subsections to the **Results** section, so we thank the reviewer for their valuable insight. We have improved our story and changed the manuscript to incorporate molecular changes suggested (mutations, amplification or deletions, DNA methylation; we also added chromatin accessibility).

We chose molecular changes that explained multiple mechanisms to activate oncogenes (OCGs) and inhibit tumor suppressor (TSGs).

We observed hypermethylation in the promoter regions for tumor suppressor predicted by Moonlight, whereas hypomethylation for oncogenes (**R5**). For example Moonlight identified GAS7 as a hypermethylated tumor suppressor in lung cancer [PMID: 26506240] and as an hypomethylated oncogene in head and neck squamous cell tumors. In particular GAS7 has been associated with copy number changes head and neck cancer cell lines [PMID: 20014447], but it has no been validated yet as oncogene for head and neck tumors, suggesting an interesting target for future studies.

We observed a global opening of chromatin in the promoter regions for oncogenes predicted by Moonlight. Concurrently, chromatin was more closed or had dampened signal for tumor suppressors (**R6**). For example Moonlight identified LSM1, a gene predicted by Moonlight as an oncogene and reported in literature as oncogene in breast cancer [PMID: 17001308], showed the highest peak in the promoter region by ATAC-seq chromatin accessibility.

We extended our analyses of Moonlight to include copy number data, which revealed amplification of oncogenes and deletion of tumor suppressors (**R7**). For example, we observed amplification of the oncogenes CCND1 [PMID: 10066068] and CCNE1 in breast cancer. Moreover, we identified deletions in tumor suppressors, such as DACT2 and TGFBR3. In addition, Moonlight predicted FOXM1 as an oncogene with associated amplification in colon adenocarcinoma and lung squamous cell carcinoma [PMID: 27162244,25561901]

We also extended our analyses of Moonlight to include mutation types, which revealed more somatic mutations for oncogenes and intron mutations for tumor suppressors (**R8**). For example ASPM and CMYA5 were predicted as novel oncogenes in breast cancer, while ERBB2 is an already well-known oncogene in breast cancer [PMID: 11156523]. Further, ST6GALNAC3 was predicted by Moonlight to be a tumor suppressor in breast cancer with 33 samples with intron mutations.

3.5. The discussion section requires more biomedical interpretation of the obtained results. Currently, there is too much focus on the descriptive abilities of the Moonlight rather than on the biological findings it delivers. The authors should highlight known dual-role genes that were identified by Moonlight that validate the methodology. On the other hand, authors should discuss more biological function of newly identified dual-role genes with potential clinical implications; e.g. these that correlate with the survival of cancer patients. Also, disease-free survival should be used for the statistical analysis, if possible.

The reviewer's comment provides a valuable perspective. We have incorporated more biomedical interpretation of the results, not only integrating multiple studies from high impact factor journals that supported our findings (*Nature, Cell, Science*), but also highlighting the novelty of our story within a larger biological- and cancer-related context. As an example, a recent publication in *Science* discussed the permissive state of tissue with pre-existent epigenetic changes, allowing a gene to be an oncogene and tumor suppressor [PMID: 30872507]. Their multiple findings agreed with Moonlight's findings. Besides this, we also reported several examples of known dual-role genes identified by Moonlight (BCL2, CDKN2A, KIT, and SOCS1) that validated the methodology for the expert-based and machine-learning approaches.

Moonlight also identified ANKRD23 (Ankyrin Repeat Domain 23) as a dual-role gene. Moonlight predicted this gene as an oncogene in renal clear-cell carcinoma associated to poor survival prognosis and a tumor suppressor in urothelial bladder carcinoma with good survival prognosis. We believe that this gene deserves further investigation, especially in regards to the immune system. We conducted the survival analyses based on overall survival, but for our future work we wish to consider disease-specific survival, disease-free interval, and progression-free interval. These endpoints are described in a recent paper [PMID: 29625055]. In this way, we will have a better comprehensive study reflecting tumor biology, aggressiveness, or responsiveness to therapy and association to cancer driver genes.

Minor comments:

The abbreviations for gene types and processes are overused and should be limited when really needed. It will improve the flow of the text and will make it less technical.

To satisfy the reviewer's request, we have removed all the abbreviations we had in the Results section of the manuscript to reduce the technical nature of the paper.

REVIEWERS' COMMENTS:

Reviewer #3 (Remarks to the Author):

I am very pleased with the revised version of the manuscript. The authors significantly improved the overall flow and logic of their story. The most critical points addressed by the reviewers have been fully addressed.

At this stage, the manuscript is fully acceptable for publication and I am positive that the publication will satisfy the broad interests of the readers of Nature Communications.